# Arctos: Community-driven innovations for managing natural and cultural history collections

Carla Cicero[1☯¶]*, Michelle S. Koo[1☯¶]*, Emily Braker[2☯¶], John Abbott[3¶], David Bloom[4¶], Mariel Campbell[5¶], Joseph A. Cook[5,6¶], John R. Demboski[7¶], Andrew C. Doll[7¶], Lindsey M. Frederick[8¶], Angela J. Linn[9¶], Teresa J. Mayfield-Meyer[10¶], Dusty L. McDonald[10¶], Michael W. Nachman[1¶], Link E. Olson[9¶], Dawn Roberts[11¶], Derek S. Sikes[9,12¶], Christopher C. Witt[5,6¶], Elizabeth A. Wommack[13¶]

1 Museum of Vertebrate Zoology, University of California, Berkeley, California, United States of America, 2 University of Colorado Museum of Natural History, University of Colorado, Boulder, Colorado, United States of America, 3 Department of Museums Research and Collections and Alabama Museum of Natural History, The University of Alabama, Tuscaloosa, Alabama, United States of America, 4 VertNet, Sebastopol, California, United States of America, 5 Museum of Southwestern Biology, University of New Mexico, Albuquerque, New Mexico, United States of America, 6 Department of Biology, University of New Mexico, Albuquerque, New Mexico, United States of America, 7 Denver Museum of Nature & Science, Denver, Colorado, United States of America, 8 New Mexico Museum of Natural History & Science, Albuquerque, New Mexico, United States of America, 9 University of Alaska Museum, University of Alaska Fairbanks, Fairbanks, Alaska, United States of America, 10 Arctos Consortium, Oakland, California, United States of America, 11 Chicago Academy of Sciences, Chicago, Illinois, United States of America, 12 Department of Biology & Wildlife, University of Alaska Fairbanks, Fairbanks, Alaska, United States of America, 13 University of Wyoming Museum of Vertebrates, University of Wyoming, Laramie, Wyoming, United States of America

☯ These authors contributed equally to this work.
¶ Authors are current or former Officers, Board of Director members, or staff for the Arctos Consortium
* ccicero@berkeley.edu (CC); mkoo@berkeley.edu (MSK)

**Data Availability Statement:** The data underlying the results presented in the study are available from the Arctos Consortium through the public Arctos data portal (https://arctos.database.

## Abstract

More than tools for managing physical and digital objects, museum collection management systems (CMS) serve as platforms for structuring, integrating, and making accessible the rich data embodied by natural history collections. Here we describe Arctos, a scalable community solution for managing and publishing global biological, geological, and cultural collections data for research and education. Specific goals are to: (1) Describe the core features and implementation of Arctos for a broad audience with respect to the biodiversity informatics principles that enable high quality research; (2) Highlight the unique aspects of Arctos; (3) Illustrate Arctos as a model for supporting and enhancing the Digital Extended Specimen concept; and (4) Emphasize the role of the Arctos community for improving data discovery and enabling cross-disciplinary, integrative studies within a sustainable governance model. In addition to detailing Arctos as both a community of museum professionals and a collection database platform, we discuss how Arctos achieves its richly annotated data by creating a web of knowledge with deep connections between catalog records and derived or associated data. We also highlight the value of Arctos as an educational resource. Finally, we present the financial model of fiscal sponsorship by a nonprofit organization, implemented in 2022, to ensure the long-term success and sustainability of Arctos.

museum). All development code is available through the Arctos repository on GitHub (https://github.com/ArctosDB/arctos). For more information, contact the Arctos Working Group Executive Officers (arctos-working-group-officers@googlegroups.com).

**Funding:** Funding for development and sustainability of the Arctos collection management system has been supported by grants from the National Science Foundation, including DBI-9630909, DBI-9876837, DBI-2034568, DBI-2034577, DBI-2034593, DEB-9981915.

**Competing interests:** The authors have declared that no competing interests exist.

# 1. Introduction

Museums house millions of objects and associated data records that document biological, geological, and cultural diversity across different spatial and temporal scales. In recent decades, digitization efforts have greatly increased accessibility to these data, unleashing exciting initiatives in research and education [1–6] and revolutionizing interdisciplinary studies in evolutionary biology, biogeography, epidemiology, cultural change, and human-mediated environmental impacts [2,7–9]. Furthermore, community science efforts aimed at digitizing museum data have shown that diverse communities can be engaged in enhancing the scientific value of museum collections [10]. While collections data are increasingly available, end users producing research, educational tools, and policy derive the most benefit from carefully curated, high-quality, discoverable information that comprehensively assembles and links everything that is known about objects in an extended network [11–13].

Collection information management systems range from simple spreadsheets to sophisticated relational databases. Fortunately, advances in informatics focused on natural and cultural heritage collections have enabled broad-scale aggregation of museum data [14] from different sources through the development of global metadata standards (e.g., Darwin Core, https://dwc.tdwg.org; Dublin Core, https://dublincore.org; Getty Vocabularies, https://www.getty.edu/research/tools/vocabularies). Although these initiatives have massively increased the quantity of data that are available, the quality of data depends strongly on local controls that standardize and improve the consistency of data values [11]. Such efforts to standardize data are greatly improved by user input, especially when diverse disciplines with varying perspectives are represented [15]. Likewise, principles for scientific data management and stewardship rely on structured information to promote data discovery and use through transparency, reproducibility, and reusability [16].

Increasingly, museum curators and collection managers are faced with multiple considerations and challenges when selecting from different systems to manage and publish collections data. While many platforms exist, not all collection management systems have the advanced infrastructure needed to integrate diverse datasets and examine complex interactions and processes [17]. A 2021 survey conducted by iDigBio summarizes and compares 11 collections management systems available for natural history museums [18]. Here we describe one such system, Arctos (https://arctos.database.museum), in greater detail. Premised on the core principles of standardization, flexibility, interdisciplinarity and connectivity, Arctos was established as a collaborative model to harmonize heterogeneous data and address novel needs and information types in response to changing technology, workflows, ethical considerations, and regulations. Nearly thirty years after implementation, its community-guided developments and active user base demonstrate that Arctos provides an effective solution for managing, linking, and publishing research-quality, complex natural and cultural history data.

## 2. A brief history of Arctos

The foundation of Arctos began in 1996 when the Museum of Vertebrate Zoology (MVZ) at the University of California Berkeley developed an information management model for its collections ("MVZ Database Model", [19]). This model was unique at the time in its ability to integrate and manage data from multiple collection types in a single environment, to relate cataloged objects across different collections, and to track and promote access to researchers and educators. The model was implemented as a web-based system and renamed Arctos in 1999 at the University of Alaska Museum (UAM) as part of the Arctic Archival Observatory. The University of New Mexico Museum of Southwestern Biology (MSB) and the MVZ began using this Arctos platform in 2003 and 2008, respectively. In 2009, a separate Arctos installation

(MCZBase) was established as a central repository for collections data at the Museum of Comparative Zoology at Harvard University. The Texas Advanced Computing Center at the University of Texas, Austiin (TACC, https://tacc.utexas.edu) began hosting media for Arctos in 2008 and has been our infrastructure partner since 2012. Through its high-performance computing servers, TACC provides a suite of core services including web and database hosting, backups of data and software, a testing environment, cybersecurity, and media storage.

Since its inception, the number of records in Arctos has increased ~8-9% annually (Fig 1A). Records are globally distributed (Fig 1B), representing objects collected from almost every country and all continents and oceans. Although originally developed for U.S. vertebrate collections, the shared system now includes over 5 million diverse objects held in 330 collections by 57 institutions from 7 countries (China, Ecuador, Ireland, Japan, Madagascar, Portugal, and USA; Fig 2, Table 1); further international expansion is anticipated and welcome. The University of Alaska Museum was the first institution to add non-vouchered observations (e.g., beached whale sightings) in 2005 and cultural object records in 2014. Over the past two decades, the Museum of Southwestern Biology has incorporated time-series samples from zoos [20] and endangered species captive-breeding programs (U.S. Fish and Wildlife Mexican Wolf Recovery Program, https://arctos.database.museum/project/1000071), as well as mark/recapture and long-term studies for ecological research networks (e.g., NEON, https://www.neonscience.org; LTER, https://lternet.edu) and emerging pathogen surveillance [21]. More recently, the Museum of Vertebrate Zoology has cataloged radio-tracked bird sightings and thousands of historic and modern bird observations from surveys such as the Grinnell Resurvey Project [22,23]. The Arctos software development model of "release early, release often" means that it responds quickly as research or collection management needs arise within the consortium. Furthermore, this flexibility opens the door for incorporating new record types such as living collections, ecological monitoring stations, and natural history reserves.

## 3. Methods: Core features and implementation of Arctos

Over time, Arctos has grown into a cutting-edge collection management system with a full suite of features for governing, hosting, managing, sharing, and connecting collection object data, people, transactions, and other information relevant to collections-based research and education in a standardized way. The database consists of a mix of structured and unstructured data which can either be collection-controlled and/or community-curated (S1 Fig). Participation is available to any institution around the globe interested in contributing, managing, and disseminating collections data. Implementation is controlled by a Virtual Private Database [24] in the relational database PostGRESQL/PostGIS [25] with a Lucee-based web interface [26], which allows each collection to manage their records independently. Cybersecurity through TACC aligns with National Institute of Standards and Technology (NIST) protocols [27,28] and with the requirements of the University of Texas, Austin. TACC also is audited by a third-party independent auditor on a fixed schedule to assess compliance with NIST. Further protection mechanisms for Arctos include data redundancy, multi-system replication, weekly backups, and plans to implement geographic redundancy through data mirroring. The Arctos application software is released under the open-source license Apache 2.0 [29] and may integrate open-source code, but its development model is not strictly open-source due to its highly specialized data and platform. External development contributions are welcome but require close collaboration for the most effective integration. All development code is available through the Arctos repository on GitHub (https://github.com/ArctosDB/arctos).

The flexibility of Arctos, combined with its aim to display all that is known about a collection object through its integrated data ecosystem, provides a rich platform for scientific and

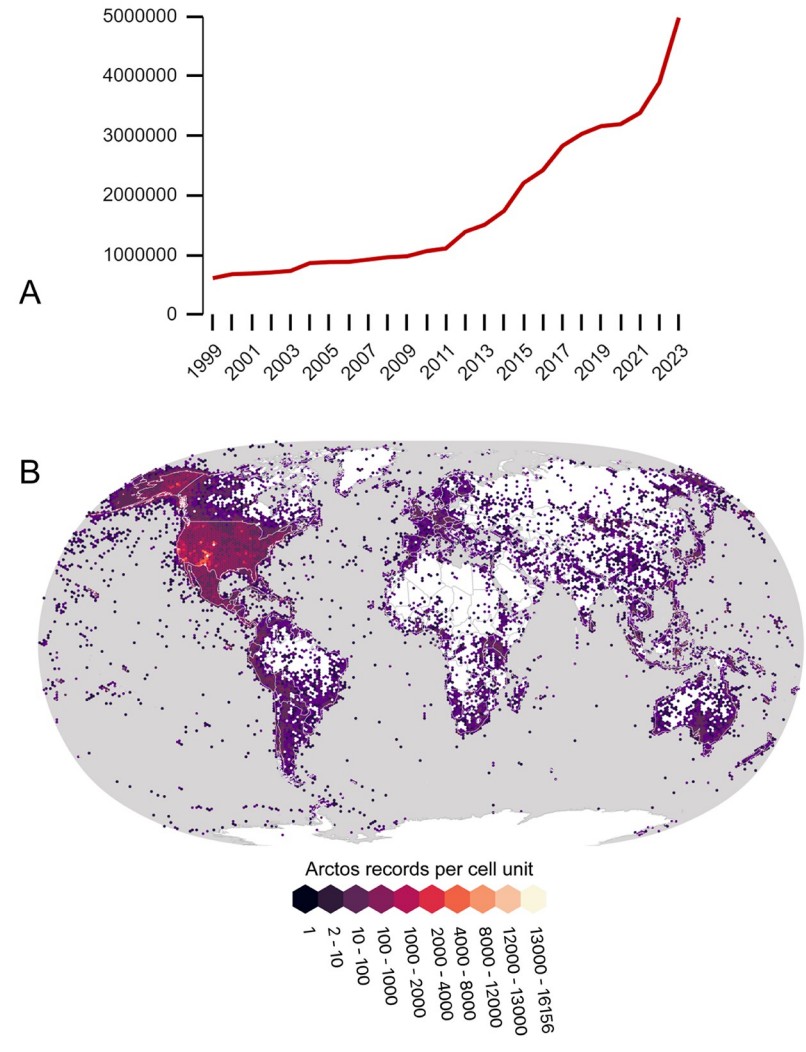

**Fig 1. Growth and geographic distribution of Arctos records.** (A) Growth of data in the Arctos Consortium showing total number of cataloged records by year (1999-2023). Arctos has grown from ~614K records in 1999 to over 5 million records in 2023. (B) Global map of georeferenced localities in Arctos per 100 km2 grid, showing 645,860 spatially distinct localities for over 3.7 million georeferenced records worldwide.

cultural discovery. Implementation of Arctos components can be categorized into three core areas (Table 2) that are described more fully below: (1) Community; (2) Features; and (3) Connectivity.

## 3.1 Community

The Arctos community is a collaborative, self-governing consortium of over 270 collection-based operators (museum directors, curators, collection managers, technicians, information specialists, students, and volunteers) from diverse disciplines and institutions. These

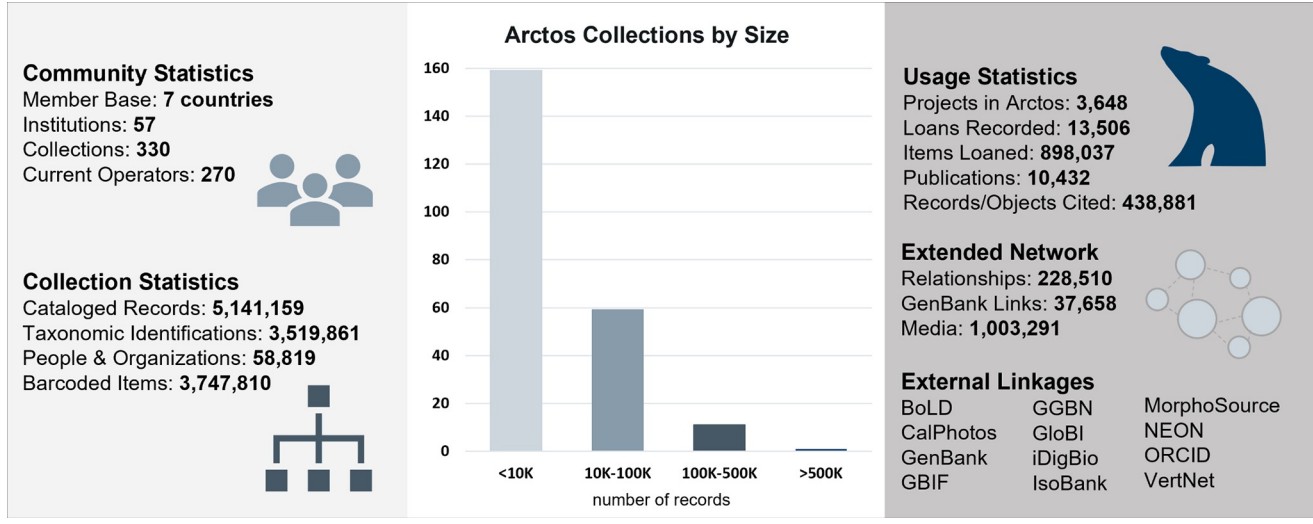

**Fig 2. Snapshot of Arctos system statistics.** Arctos system statistics across all collections as of January 2024. Additional statistics are available to users with operator permissions.

individuals are directly responsible for day-to-day database operations for their collections such as entering and updating data, processing transactions, tracking usage, reviewing issues, and/or managing administrative details. Within this broader community, members are organized into two main groups that are responsible for the overall governance and operations of Arctos (Fig 3): a small Advisory Committee that functions as the Arctos board of directors and has fiduciary and strategic responsibilities, and a larger Working Group composed of Executive Officers, a Members Council, and a paid Technical Team. The Arctos Working Group meets online at least twice monthly to discuss and prioritize issues, and focused subcommittees meet monthly or more often as needed. These meetings provide a forum for community members to raise specific questions, provide comments, and discuss concerns. The Members Council is open to all collections, although participation is not required. Such commitments are flexible in recognition of time burdens on collections staff (i.e., members can participate as much as they have time for, or only for topics of special interest). In addition to sharing knowledge and expertise, members of the Arctos Working Group drive development priorities and

**Table 1. Diversity of biological, geological, and cultural record types held in Arctos collections.**

| Records | • Archaeological objects and zooarchaeological specimens |
|---|---|
| | • Archives (field notes, ledgers, correspondence, other documents) |
| | • Artwork |
| | • Entomological, invertebrate, and vertebrate specimens |
| | • Environmental samples |
| | • Ethnological and historical objects |
| | • Genetic and genomic resources |
| | • Geological and mineralogical materials |
| | • Herbarium specimens |
| | • Living organisms |
| | • Meteorites |
| | • Microbiological samples |
| | • Observational occurrences |
| | • Paleontological specimens |
| | • Parasitology specimens |
| | • Physical and digital media |
| | • Teaching materials |

**Table 2. Overview of the core features of Arctos.**

| Community | • Active forum for community discussion of needs and priorities<br>• Advisory Committee (oversight board) with fiduciary and strategic responsibilities<br>• Collaborative training, documentation, and proposals<br>• Executive Officers who oversee daily operations<br>• Members Council of collection representatives guiding development<br>• Network of diverse expertise in managing different types of collections<br>• Peer mentorship for new and existing collections<br>• Regular meetings to discuss issues and community needs<br>• Shared vocabularies and authorities to improve data consistency and retrieval |
|---|---|
| Features | • Attributes for records, localities, events, and agents<br>• Automated reminders and data-quality checks<br>• Batch edits and updates of existing records<br>• Collection-level control over data management, permissions, access<br>• Creation and management of projects, publications, and citations<br>• Data entry and bulkloading of new records<br>• Data migration and cleaning services<br>• Export of records in standards-compliant formats (e.g., csv, eml, kml, txt)<br>• Flexible identifiers for connecting different records<br>• Generation of labels, reports, and invoices<br>• Internal statistics and data quality notifications<br>• Management of information about people and organizations<br>• Management of taxonomy and identification history<br>• Management of transactions (accessions, loans, borrows, permits)<br>• Object tracking and machine-readable labels integrated with catalog records<br>• Semi-automated georeferencing, reverse geocoding, and mapping |
| Connectivity | • Authorities and services for agents and organizations<br>• Data publishers and aggregators<br>• Datasets on biological interactions<br>• Digital Object Identifiers for publications<br>• Project management and development platforms<br>• Repositories for genetic, isotopic, and morphological data<br>• Resources for taxonomy, nomenclature, and conservation status |

community policies, engage in outreach, produce documentation, form a network of peers that are available to mentor new and existing collection representatives, and create community-generated training modules (e.g., tutorials, https://arctosdb.org/learn/tutorial-blitz;en webinars, https://arctosdb.org/learn/webinars; documentation, https://handbook.arctosdb.org).

Community discussion focuses primarily on Arctos data that are shared among all collections, e.g., authorities and controlled vocabularies pertaining to taxonomy, geography, preparations, attributes, and agents (people and organizations) (S1 Fig). Data standardization is a core tenet of Arctos, and database features and enhancements are forged from input among Arctos users. New features or controlled vocabulary values requested by one collection and approved by the community are accessible to all users, resulting in a benefit to the community as a whole. Learning how to use and contribute to Arctos also is a collaborative process. This is especially useful to personnel who are less experienced or who are interested in expanding their knowledge of collection management tools. Although consensus-building presents its own challenges, the Arctos model is highly responsive to emerging innovations and collection needs by encouraging community-based solutions, workflow efficiencies, and data quality improvements – thereby advancing best practices in collection data management and data fitness for use [11,30].

Overall, the Arctos community's goal is to make research-grade collection data discoverable through an openly accessible and richly networked online portal (https://arctos.database. museum). This portal serves to promote multidisciplinary research and public understanding of natural and cultural history by integrating datasets across biological, geological, and cultural collections. In addition, it supports and enhances the mission of government agencies at all

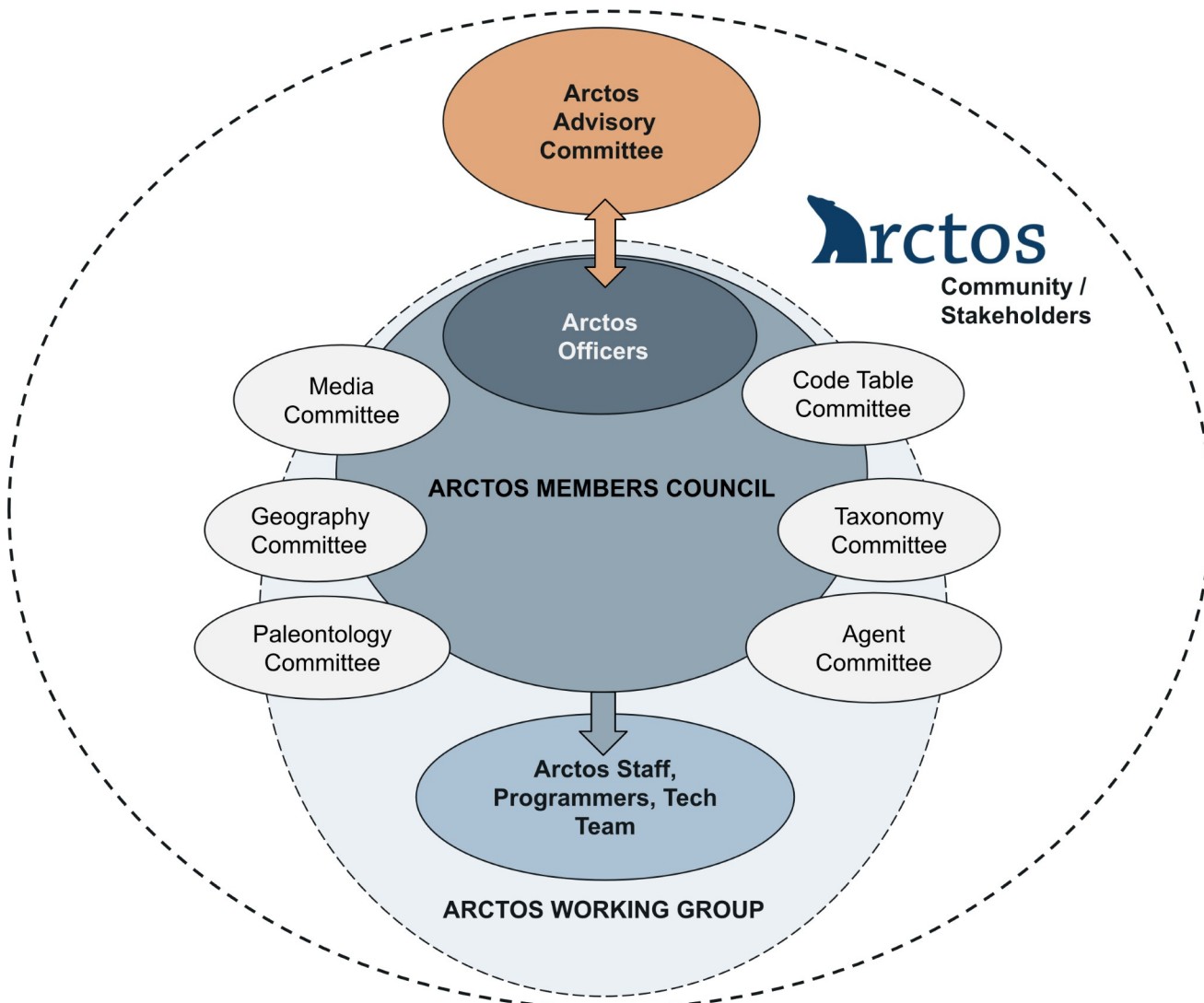

**Fig 3. Arctos community organization chart.** Within the Arctos community, the Working Group is the primary organizational body that oversees policies, procedures, and implementation. This group is composed of three core elements: (1) Executive Officers, elected volunteers who oversee communications and general functioning of the Arctos community; (2) a Members Council made up of volunteer collections representatives who participate in officer elections, community practice discussions, and technical subcommittees focused on particular database functions; and (3) a paid Technical Team that includes a lead programmer, database administrator, and community coordinator, as well as contract programmers as needed. The director of the Executive Officers generally organizes Arctos Working Group meetings, supervises staff, and oversees daily operations. See https://arctosdb.org/contacts for a list of all board members and officers.

levels by serving as a repository of curated data for items collected on publicly owned lands (e.g., parks and reserves, land management areas, wildlife refuges, etc.). Although the majority of such records are in the U.S., there are no geographic or administrative restrictions.

## 3.2 Features

Arctos is an entirely web-based platform such that members do not need to spend time or financial resources installing or updating software, maintaining servers, responding to security threats, or coordinating backups. Because the objects and data are curated by different institutions, Arctos also functions as its own data aggregator and publisher. Collection management

tools are the nuts and bolts of Arctos functionality, enabling data entry, editing, and searching as well as improving data quality and increasing discoverability. Users can correct or update records by either editing single records or by using batch upload tools. Each cataloged record has a Globally Unique Identifier (GUID) that is formatted as a DarwinCore Triplet (institution:collection:catalognumber) and is part of the Arctos URL (e.g., https://arctos.database.museum/guid/APSU:Fish:1079). In addition to identifying records within Arctos, these links are shared with external users, platforms, or aggregators. Data are exported in a standards-compliant format compatible with spreadsheets, online mapping platforms, and other services. Arctos-wide and collection-specific statistics provide data summaries that are useful metrics for understanding the scope and status of collections data as well as for meeting annual reporting requirements. Finally, Arctos reports generate printable transaction invoices, collection ledgers, and object labels. Report templates are shared across the community and can be modified to institutional specifications.

Access to different features depends on a user's training and roles. For example, students and volunteers may be given access to enter data for a specific collection by choosing values from existing controlled vocabularies, but they are typically restricted from changing information such as taxonomy or geography that are shared among all collections. The addition of new values or changes to existing controlled vocabularies can be proposed by any Arctos member but require community approval; editing of controlled vocabularies is limited to focused Arctos committees. This permissions hierarchy limits misspellings and duplication of values (e.g., the same person entered multiple ways), ensures that entries are verified (e.g., a country, state, or county is valid), and compels consistent values for certain fields (e.g., sex, preservation method). Ultimately, the focus on standardizing data values leads to higher data quality [11] and increases discoverability for researchers and educators using specific search criteria.

Below we describe how the following core features operate in Arctos: data entry and encumbrances/embargoes; attributes; agents and identifiers; taxonomy and identifications; transactions; projects and publications; object tracking; and spatial data quality. We highlight these features for their importance in the management and use of natural and cultural history collections.

**3.2.1 Data entry and encumbrances/embargoes.** New records are entered individually or by batch uploading (aka bulkloading), validated through a series of Arctos data checks, and accepted by curators or collection managers prior to loading into the online public database. The single-record data entry form is highly customizable, and users can save their modified layout which streamlines curatorial workflows. Likewise, the Arctos "bulkloader builder" function allows users to create a customized comma-separated file for batch uploading, whereby they can specify the type and number of fields to include. Arctos can integrate digitization workflows, such as creating records and capturing metadata from images of collection objects and their label text, as was done with the University of Alaska Museum's herbarium collection. Use of code tables and controlled values standardizes entries across all collections. However, database operators can customize their own collection settings for certain controlled fields such as part names; for example, users entering bird data will see different part names than those entering insect data, and each collection can choose which parts to use based on specific preparation practices. Once collections data are uploaded, they are immediately accessible online unless a data operator chooses to encumber or embargo the information. Encumbrances and embargoes protect sensitive data such as sacred cultural information, collecting locations for fossils or endangered species, and archaeological resources. The Arctos community has developed guidelines for data redaction measures as required by paleontological and cultural collections to meet U.S. federal and private land regulations (e.g., Paleontological Resources Preservation Act of 2009, 16 U.S.C. § 470aaa 1-11; National Historic Preservation

Act of 1966, Public Law 89-665; 54 U.S.C. 300101 *et seq;* and Archaeological Resources Protection Act of 1979, 16 U.S.C. 470aa-470mm; Public Law 96-95 and amendments). Likewise, encumbrances and embargoes restrict usage of collection objects and data according to deposition agreements (e.g., delayed public access until research publication), permit requirements, material transfer agreements, or other regulatory instruments (e.g., Nagoya Protocol [31]).

**3.2.2 Attributes.** Attribute fields are used throughout Arctos to record descriptive characteristics about cataloged items and associated data fields such as localities. This can be object-level information such as standard trait data routinely recorded in natural history disciplines (e.g., measurements, life stage, conductivity units), or metadata related to managing art and ethnographic objects (e.g., subject, cultures, copyright status). Attributes housed in the locality table are used to capture place-based information (e.g., drainages, lithostratigraphy data, landholder), while the event table stores temporal attribute data (e.g., air or water temperature, relative humidity, weather). Identification attributes allow users to record information pertaining to the nature and confidence of a determination, while part attributes add context to each component of the cataloged item (e.g., preparation or preservation method).

The format for entering attributes varies from a controlled vocabulary to free-text or integer, and values are accompanied by a suite of determination fields where attribute determiner, date, and method can be documented, allowing for multiple assertions that are all trackable. Critically, attributes encourage data managers to structure data for powerful and precise queries, such as searching for all records of adult rodents that have liver tissue available and have been examined for *Orthohantavirus* (Fig 4). Controlled values are managed by a Code

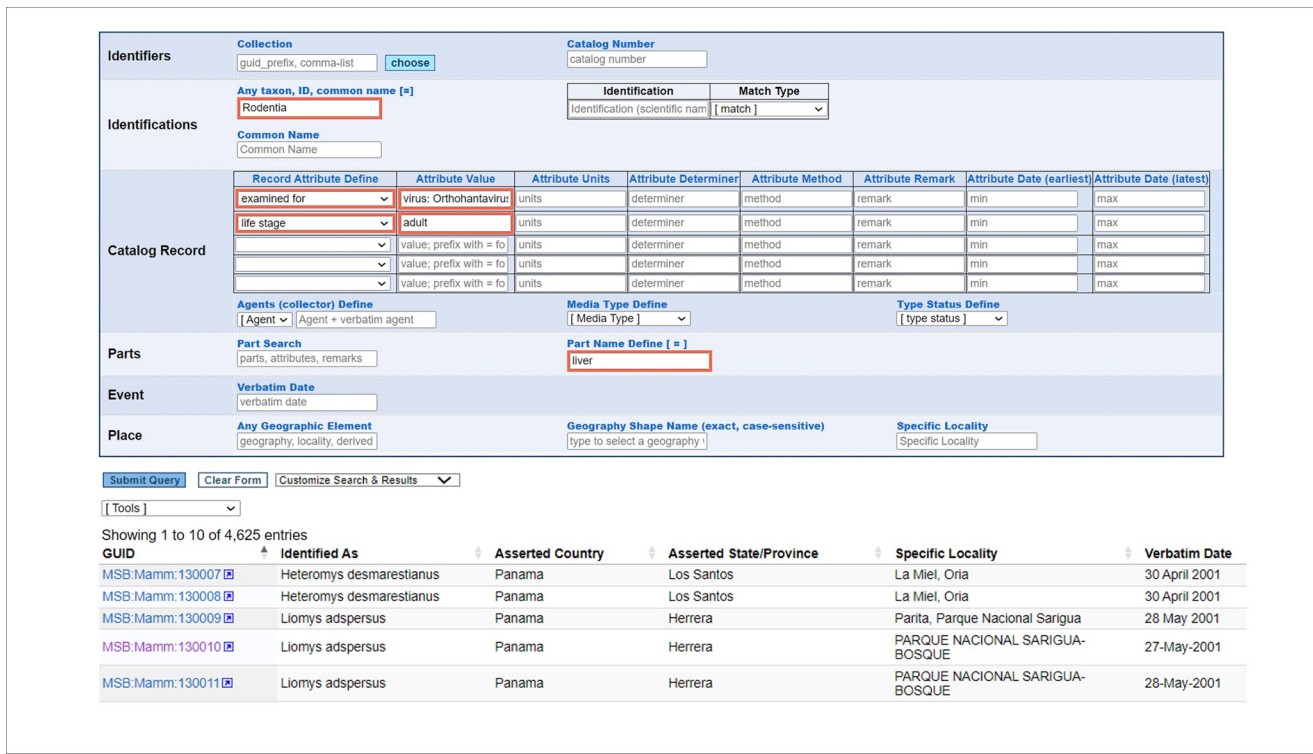

**Fig 4. Multi-parameter search.** The search pane allows users to enter multiple query parameters that can be customized. A search for adult rodent (order Rodentia) specimens with liver tissue that have been examined for *Orthohantavirus* yields 4,625 results (27 Mar 2024). This query can be replicated using the Saved Search link https://arctos.database.museum/saved/Multi%2Dparameter%20search, which saves search parameters as a shareable URL that is updated dynamically as new records fitting those parameters become available.

Table Committee composed of data managers from across the Arctos community that collaborate with the Database Administrator to evaluate new requests. This process involves verifying that suggested values are suitable, well-defined and/or linked to an accepted authority, and broadly usable by the most collection types possible to aid in Arctos-wide data standardization and discovery. Certain types of attributes such as for traits are specific to the collection type (e.g., "carapace" applies to herpetology and earth science but not to entomology collections), but it is easy to apply existing values to a new collection type so that they are available for use. Examples of two attribute code tables are accessible at the following URLs: https://arctos.database.museum/info/ctDocumentation.cfm?table=ctattribute_type; https://arctos.database.museum/info/ctDocumentation.cfm?table=ctlocality_att_att). All Arctos code tables are available at https://arctos.database.museum/info/ctDocumentation.cfm.

**3.2.3 Agents and identifiers.** Arctos provides detailed information about people, organizations, and institutions, referred to as 'agents,' including alternate names (a preferred name plus any number of "aka" names), relationships (e.g., "student of" or "associate of"), birth and/or death dates if known, address history, associated unique identifiers (e.g., ORCID, wikidata IDs), biographical information available to the public, and activities associated with Arctos records (Fig 5). Any contact information (e.g., emails, phone numbers, addresses, etc.) is only accessible to curatorial users and is not public.

Agents are integrally tied to identifiers that are applied to records through a combination of identifier type (e.g., catalog number, collector number, lot number) and the person, organization, or institution that issued the identifier. A single record can have multiple identifiers which provides flexibility when capturing legacy collection practices. Thus, collections that employ an integer catalog number may record and search previously associated catalog

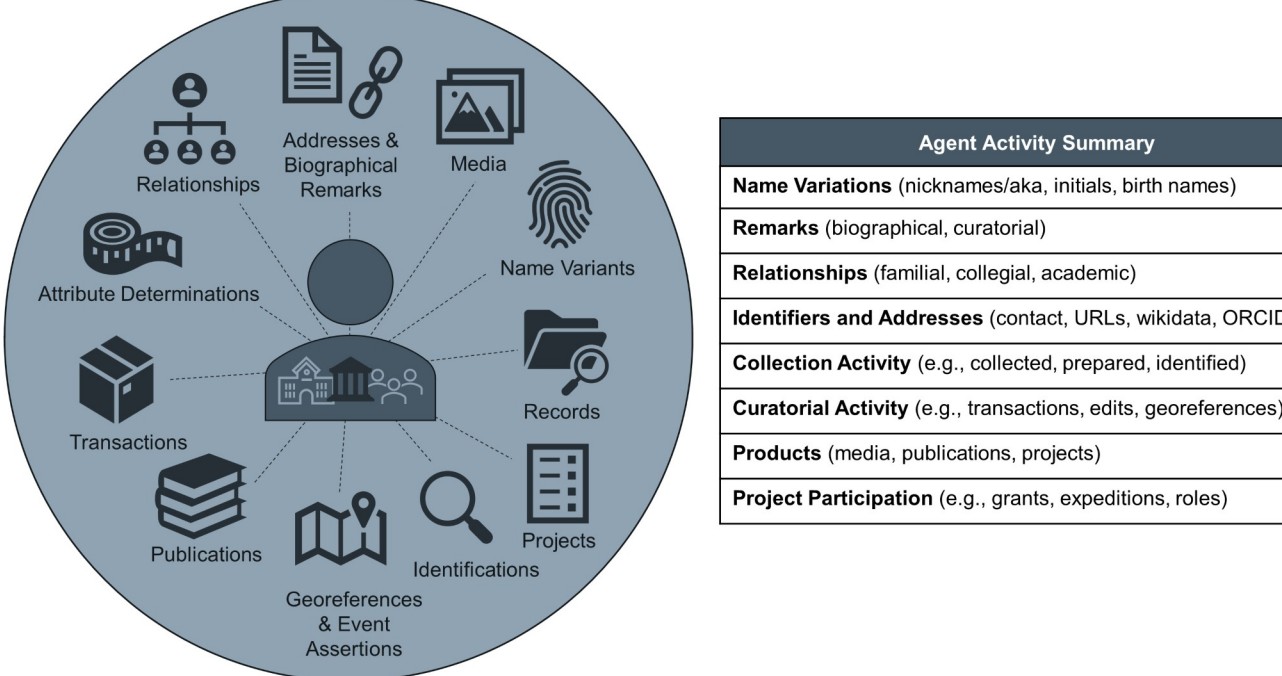

**Fig 5. Synopsis of agent information and activity in Arctos.** Arctos provides a holistic view of research and curatorial contributions by people, organizations, and institutions (i.e., "agent activity") to facilitate data relationships, attribution, and assessment metrics. Dynamic links associate agents with related agents (e.g., academic lineages, family members), external resources (wikidata, ORCID), collection activities (objects collected, prepared, or identified), curatorial work (transactions, edits, georeferences), and projects and products (publications, media, grants, expeditions).

numbers that used a different format or were historically cataloged in a separate collection. Furthermore, identifiers are used to establish relationships between related records such as host-parasite or parent-offspring. Finally, Arctos can integrate any resolvable identifier and accommodates URLs that link out to related records, either within Arctos or externally to providers such as GenBank for genetic or genomic data (over 37,600 GenBank links as of January 2024).

**3.2.4 Taxonomy and identifications.** Identifications within Arctos are treated separately from taxonomy. Taxonomy refers to formal classification systems, including cultural lexicons, and Arctos allows for different classification systems using external sources (e.g., Global Names Architecture [32]; Nomenclature 4.0 [33]; World Register of Marine Species [34]) as well as customized taxonomies. This provides both data quality control and the flexibility of collections to choose and modify their own nomenclature. Collections staff apply names to objects through a flexible identification module that allows for vernacular, regional, and Indigenous names, taxonomic uncertainties, biological realities (e.g., hybrids, intergrades, new species with temporary designations), multiple identifications, non-hierarchical identifications, and nomenclature from geological, archival, art, and cultural collections. For example: a hybrid specimen of white mulberry (*Morus alba*) and red mulberry (*Morus rubra*) is identified by selecting both parental names from the Arctos taxonomy table (e.g., https://arctos.database. museum/guid/CHAS:Herb:2021.10.4); a parka made from fur of multiple mammal species is identified taxonomically by the cultural and biological names of its components (e.g., parka, *Bos taurus*, *Canis lupus*, *Gulo gulo*, *Rangifer tarandus*; https://arctos.database.museum/guid/ UAM:EH:UA91-014-0001, Fig 6); and a Song Sparrow (*Melospiza melodia montana*) nest containing the eggs of a brood parasitic cowbird species (*Molothrus ater artemisiae*) is identified by the taxonomic names of both the host and the parasite (https://arctos.database.museum/ guid/MVZ:Egg:609). Identification updates are applied to single records or in batches, critical for the management of collections that need to regularly update identifications of large numbers of specimens. Importantly, Arctos records the history of all changes made to identifications. When taxonomy is revised or new identifications and associated metadata (e.g., determiner, date, basis of determination) are added, old identifications are retained and remain searchable. These may include erroneous identifications, taxonomic synonymies, or sequential identifications based on close diagnosis of features (e.g., a rove beetle, *Placusa tacomae*, was identified to family by a naturalist in 2009, then to subfamily by an expert in 2014, and finally to species by an expert of that subfamily in 2015; https://arctos.database.museum/ guid/UAM:Ento:110123).

**3.2.5 Transactions.** Collections' transactions including accessions, loans (outgoing collection material), borrows (incoming material from another collection), and permits are all managed in Arctos. Accessions, loans, and borrows are collection-specific, with options for formatting transaction numbers depending on in-house curatorial practices. Permit metadata can be linked to accessions, loans, and borrows for compliance with state, federal, and international (e.g., CITES, Nagoya) regulations [31]. In addition to capturing basic information about the transaction (i.e., persons and/or agencies involved, date, transaction number, nature and amount of material, remarks), transactions can be linked to images or documents that provide supporting information. Furthermore, user requests for data records or object photographs (rather than for the corresponding objects per se) are tracked through data loans and media loans, respectively. Together, Arctos transactions allow for comprehensive documentation about how records, information, and associated elements are being used.

**3.2.6 Projects and publications.** A special feature of Arctos is the ability to link transactions to thematic and dynamically linked public web pages called "projects" that summarize the contributions and uses of cataloged records for different activities (e.g., field expeditions,

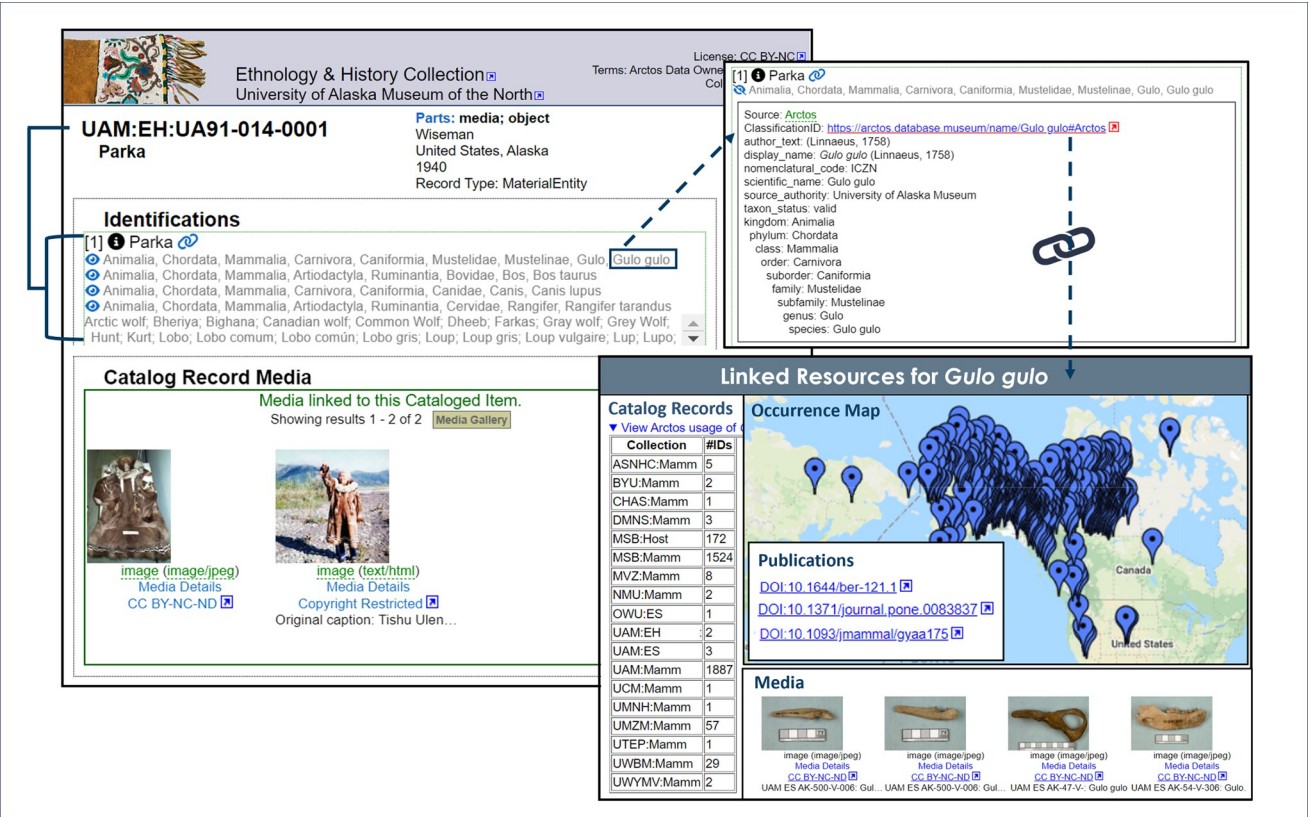

**Fig 6. Example of an Arctos record that has both cultural and biological information in its identification.** Arctos record from the University of Alaska Museum Ethnology and History Collection (https://arctos.database.museum/guid/UAM:EH:UA91-014-0001) showcasing a cultural object composed of biological materials. The object is identified with all component taxonomic names. Expansion of each taxon name in the identification provides classification details as well as linked Arctos resources such as mapped localities, media, and other records that use the same identification across collections and institutions.

digitization initiatives, collaborative research, exhibitions, collections on state or federal lands). Projects also document biosampling contributions by Indigenous communities for resource policy and decision-making as well as highlight cultural object acquisitions, displays, and practices. Projects linked to accessions and loans automatically update as records are added to the transaction, and publications and media resulting from these activities are easily associated with the project page. Publications may be cross-referenced to Digital Object Identifiers (DOIs) and cataloged records may be linked via citations, directly showing the usage of individual records or objects within or across collections. Projects are automatically related to other projects based on shared objects, enabling deep-tracking of a collections' utility and products through time. Projects are especially useful for providing agencies and funders with up-to-date information for specific activities. Thus, projects serve as the central hub that links all related data and highlights how researchers, educators, organizations, and others are using Arctos for discipline-specific or interdisciplinary goals (Fig 7, Table 3).

**3.2.7 Object tracking.** Object tracking utilizes a barcode and label system to track materials from the time of collection and accession through cataloging, storage, and loans. For example, genetic resource collections can be organized in a hierarchy that associates cataloged parts with barcoded cryovials in specific positions in boxes, racks, freezers, and rooms within buildings (e.g., nested "containers"; Fig 8). Similarly, object tracking may be used to track the locations of dry specimens in cabinets or fluid-preserved specimens in jars on shelves. Arctos can

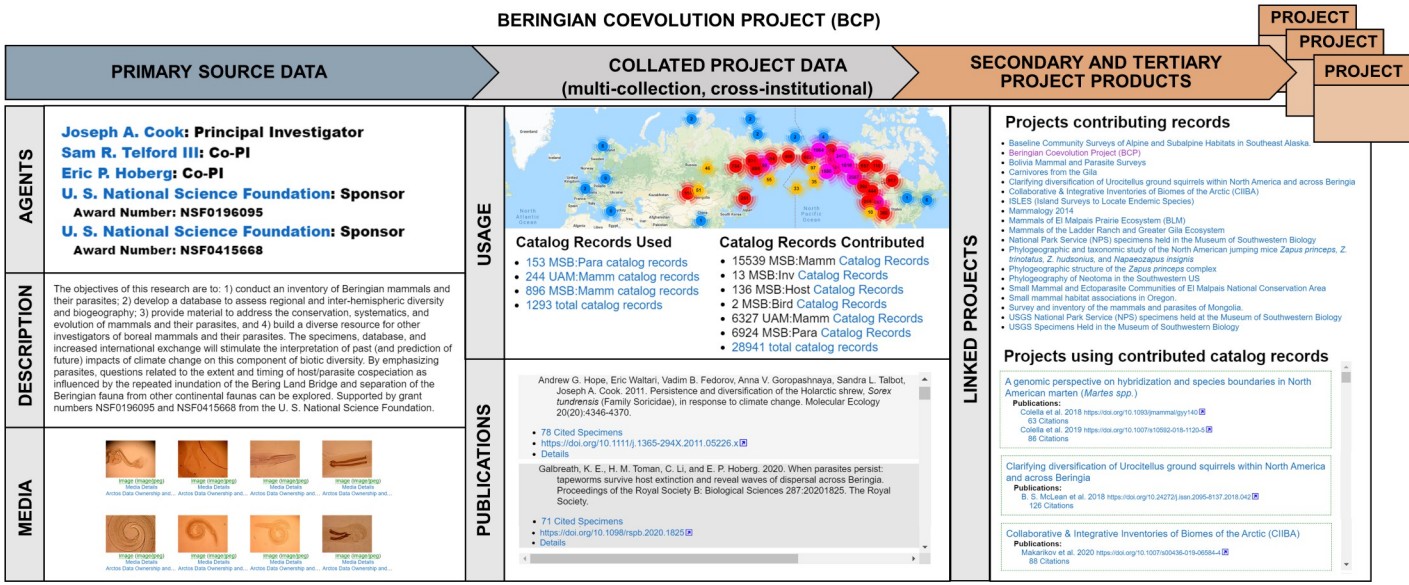

**Fig 7. Synopsis of the Beringian Coevolution Project in Arctos.** The Beringian Coevolution Project (https://arctos.database.museum/project/51) showcases primary source records and a long list of derivative products. The project includes 28,941 cataloged records representing co-examined boreal mammals and associated parasites from two museums and has resulted in 8 media objects, 216 publications citing 8,561 specimens, and a dynamic web of 153 related projects that either contributed records used by this project or that used specimens from this project to generate additional research outputs (293 subsequent publications with 69,087 citations).

accommodate different types of scannable codes (e.g., a true barcode) or non-scannable codes (e.g., cabinet numbers printed on labels), and these can be used to locate both cataloged and non-cataloged (in process) items. In addition to tracking the location of collection objects, container environments (e.g., relative humidity, temperature, ethanol concentration, pest checks) and their history can be recorded to better monitor collections and document changes over time. Object tracking in Arctos is broadly integrated with a variety of curatorial functions including data entry and quality control, accession and loan management, collection inventories and moves, conservation and preservation of collection objects, and Integrated Pest Management practices.

**3.2.8 Localities, georeferencing, and mapping.** Geography in Arctos adds critical value to records as fundamental metadata and as a measure of data quality. Integrated spatial data allow records to be correlated with environmental and geographic data, thus ensuring their usefulness beyond Arctos [35–37]. Over 75% of the more than 870,000 unique localities in Arctos have associated georeferences that are available to all Arctos users and global data aggregators (Fig 1). Collections benefit from the sharing of localities regardless of collection type (Fig 9). For example, diverse taxa such as snails, fish, and salamanders collected in the same pond or at different times from the same location can share the same locality in Arctos.

In addition to sharing locations, Arctos provides a suite of tools for mapping, describing, and visualizing spatial locations of collection objects with an emphasis on accessibility and quality. An embedded GeoLocate service (https://geo-locate.org) provides interactive web mapping that can feed decimal latitude, longitude, and uncertainty values for point locations as well as polygon vertices directly back to Arctos. Arctos also uses Google Maps services for automated data-quality checking, such as reverse geocoding to verify if coordinates are within the assigned higher geography (i.e., continent, country, state/province, county). Higher geographies are defined with polygons, and countries follow a spatially explicit authority (Database of Global Administrative Areas, https://gadm.org) which uses ISO standards (International

**Table 3. Examples of projects in Arctos that target different users.**

| Topic | Arctos Project ID[a] | Target User(s) |
|---|---|---|
| Alaskan Insect Pollinators | 10000850 | Alaska Department of Fish and Game, Researchers, National Science Foundation |
| Alaskan Plant Survey | 1000054 | National Park Service |
| Albuquerque BioPark & Museum of Southwestern Biology Specimen Repository Agreement | 10002948 | Researchers, Zoos |
| Art and Ethnology Collections purchased with funds from Rasmuson Foundation | 10003033 | Artists, Funder |
| Art Work Inspired by Life | 10002855 | Artists |
| Beringian Mammal and Parasite Coevolution | 51 | Researchers, National Science Foundation |
| Center for Disease Control Hantavirus Survey in National Parks | 10002373 | U.S. Center for Disease Control, Epidemiologists, Researchers, National Park Service |
| Educational Collaboration between Art and Biology | 10003671 | Students |
| Greenwood Wildlife Rehabilitation Center Salvaged Vertebrates (Colorado) | 10004199 | Agencies/governments (city, county, state, etc.), Researchers |
| Mexican Wolf Recovery Program | 1000071 | U. S. Fish and Wildlife Service, Conservation Groups |
| oMeso: Opening Mesoamerican Herpetofaunal Diversity to Whole Phenome Imaging | 10003438 | Researchers, Agencies, National Science Foundation |
| Resurvey of Vertebrate Communities in California | 10000047 | Researchers, Agencies, National Science Foundation |
| Seal Specimens Hunted by Native Americans in Alaska | 15 | Alaska Native Harbor Seal Commission |
| Support for Ornithology and Herpetology Collections | 10003135 | National Science Foundation |

[a] Project IDs have the base URL https://arctos.database.museum/project/ (e.g., https://arctos.database.museum/project/10000850).

Organization for Standards, https://iso.org). Polygons are not limited to higher geography but can be used to describe an object's locality instead of point coordinates. Locality attributes add descriptive terms to a place (see 2.2.2 above) and localities can be verified and locked once checked by collectors or curators to ensure data integrity. Finally, both Google Maps and the BerkeleyMapper service (https://berkeleymapper.berkeley.edu) provide data visualization and spatial exploration tools.

## 3.3 Connectivity

Arctos maximizes connectivity and information about collection records by integrating with a growing list of external repositories, authorities, and databases (Table 4). Links to some sites are reciprocal while others integrate values from an authority, and connections may be updated manually or automatically through web services. People may be associated with their own persistent digital identifier (e.g., ORCID), affiliated with their record in the Library of Congress, and linked to their profile on GitHub or other platforms. External resources for taxonomy and nomenclature cover the spectrum of record types in Arctos. In addition, Arctos classifications allow validation against different online taxonomic sources and provide information about the conservation status of species, which is important for legal compliance during transactions (e.g., whether a taxon is listed on Appendix I, II, or III of the Convention on

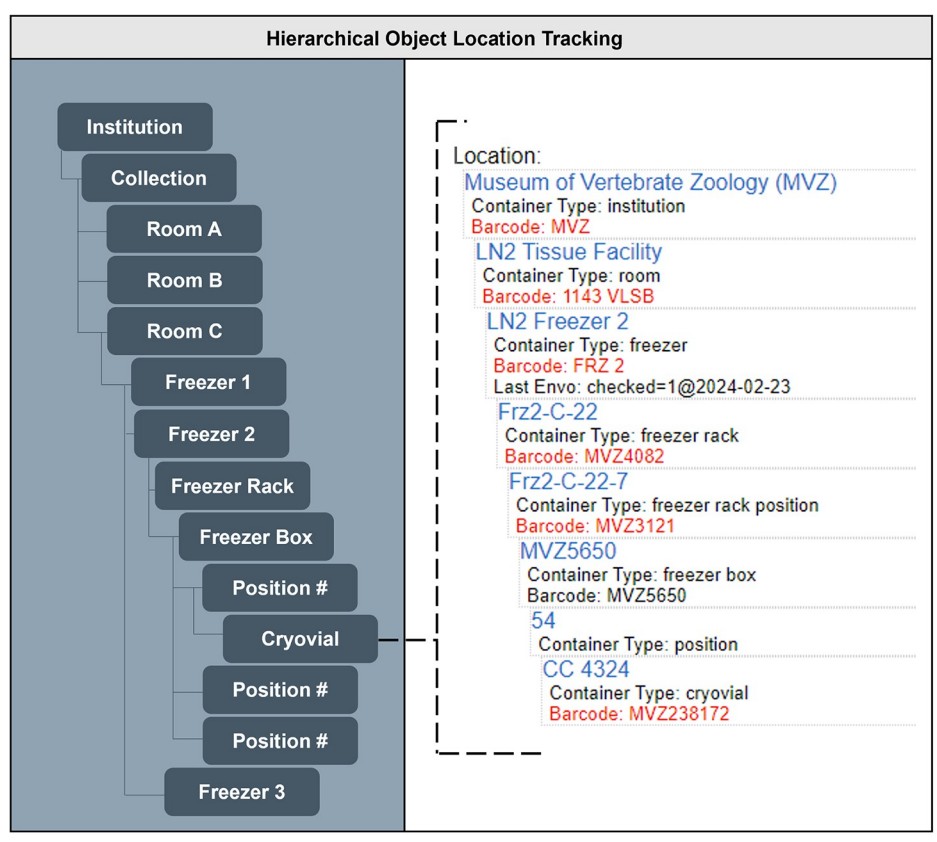

**Fig 8. Schematic showing hierarchical object tracking of tissues in Arctos.** In this schematic example, a barcoded cryovial containing tissue in the genomic resources collection of the Museum of Vertebrate Zoology (MVZ) is hierarchically nested so that its location in a freezer can be found easily. The cryovial, which is barcoded (MVZ238172) and labeled with the collector's initials and number (CC 4324), is placed in position 54 of a barcoded freezer box (MVZ5650) that is located in a specific rack, freezer, and collection room (those "containers" also have their own unique barcodes). Note that the freezer has data on the date that it was last checked as part of Arctos' environmental monitoring record.

the International Trade in Endangered Species [CITES, https://cites.org]). Arctos datasets are hosted as Darwin Core Archives on the VertNet Integrated Publishing Toolkit site (http://ipt.vertnet.org) where they are automatically published monthly, thus freeing collections staff from this often-cumbersome task, and are available for harvesting by external data aggregators including VertNet [14], the Global Biodiversity Information Facility (GBIF [38]), and the Integrated Digitized Biocollections (IDigBio) platform [4]. Data also are exported to platforms such as Global Biotic Interactions (GloBI [39]) and linked via URLs to repositories such as GenBank, IsoBank [40,41], and Morphosource [42]. Ultimately, the goal is to connect and enrich Arctos records while expanding the reach of collections for the broader community [13].

## 4. Results

For over 30 years, Arctos has expanded to serve rich data across a broad spectrum of biological, geological, and cultural collection types (Table 1). Arctos caters to collections of all sizes, with the majority holding fewer than 10K catalog records, and system statistics reflect the large community and rich data (Fig 2) that together comprise the current platform. Although the

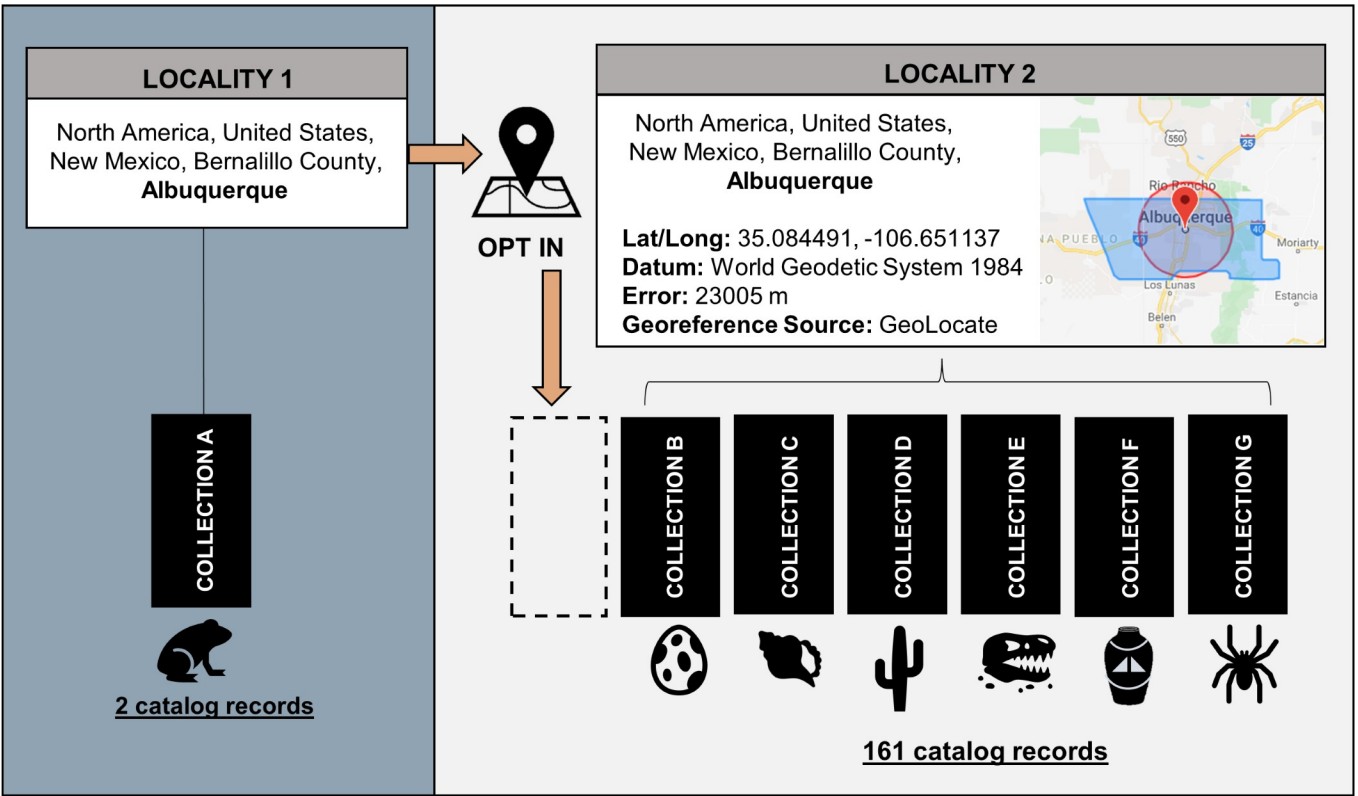

**Fig 9. Example of how localities and associated georeferences may be shared in Arctos.** Collection managers can choose to apply georeferences for a shared specific locality to their own records. In this example, the same locality is present twice in the database, with Locality 1 lacking coordinates while Locality 2 has been georeferenced. Multiple collections already share the georeferenced locality. If a collection adds records from the same place but they have not been georeferenced, that collection can opt to use the georeferenced version, thus improving their records by adding spatial data with minimal effort.

densest holdings are in North America, continued growth and especially the inclusion of more international repositories will expand geographic representation. Arctos receives heavy user traffic, with over 446K searches resulting in more than 1.4 billion records queried in 2023. Research impacts are reflected in the 10.4K publications and 435K citations associated with Arctos records.

Nearly one million media records (e.g., still images, sound recordings, 3D models, video files) are linked to Arctos data. Media may showcase objects and specimens before and after preservation [43,44] or function as a voucher for observational data. In addition, media can be key to increasing access to rare or fragile collection objects that are not typically loaned, such as eggs and nests [45,46], type specimens [47], endangered species [48], and objects that are no longer available [49,50]. Increasingly, objects are digitized in high-resolution photogrammetry and CT scanning projects [42,51,52], and those digital objects can be linked directly to Arctos records. Arctos media are not limited to documenting physical specimens or objects and observations but also portray field work, landscapes, habitats, people, and archival documents, adding value to collecting events, transactions, and agent records.

## 4.1 Access, customization, and licensing

There are multiple entry points for accessing data from the Arctos online portal depending on the query of interest. An account is not required for public access, and all records are available unless they have been encumbered or embargoed (see section 3.2.1 above). However, users

**Table 4.  Arctos connections to external repositories, authorities, and databases (as of January 2024).**

| Category | Data Service or Repository | URL |
|---|---|---|
| Taxonomy & Legal Status | AntWiki | https://www.antwiki.org |
| | Getty Art & Architecture Thesaurus | https://collectionstrust.org.uk |
| | Global Names Architecture | https://globalnames.org |
| | Hey's Mineral Index | https://www.mindat.org/cim.php |
| | Integrated Taxonomic Information System | https://www.itis.gov |
| | Nickel-Strunz Mineral Classification | http://webmineral.com/strunz.shtml |
| | Nomenclature 4.0 | https://www.nomenclature.info |
| | Species+ | https://www.speciesplus.net |
| | World Register of Marine Species | https://www.marinespecies.org |
| People & Publications | Bionomia | https://bionomia.net |
| | Crossref | https://www.crossref.org |
| | Library of Congress | https://loc.gov |
| | ORCiD | https://orcid.org |
| | PubMED | https://pubmed.ncbi.nlm.nih.gov |
| | Wikidata | https://www.wikidata.org |
| Data Repositories | Barcode of Life Data Systems | https://boldsystems.org |
| | BugGuide | http://bugguide.net |
| | CalPhotos | https://calphotos.berkeley.edu |
| | Dryad | https://datadryad.org |
| | Environmental Data Initiative | https://portal.edirepository.org |
| | GenBank | https://ncbi.nlm.nih.gov/genbank |
| | iNaturalist | http://inaturalist.org |
| | IsoBank | https://isobank.org |
| | MorphoSource | https://www.morphosource.org |
| | National Center for Biotechnology Information | https://www.ncbi.nlm.nih.gov |
| | Sketchfab | https://sketchfab.com |
| | Wikipedia | https://en.wikipedia.org |
| Data Aggregators | Alaska's Digital Archives | https://vilda.alaska.edu |
| | Biodiversity Information Serving Our Nation | https://bison.usgs.gov |
| | Consortium of Pacific Northwest Herbaria | https://pnwherbaria.org |
| | Global Biodiversity Information Facility (GBIF) | https://www.gbif.org |
| | Global Genome Biodiversity Network (GGBN) | https://wiki.ggbn.org |
| | Integrated Digitized Biocollections (iDigBio) | https://www.idigbio.org |
| | SCAN Portal | https://scan-bugs.org |
| | SEINet Portal | https://swbiodiversity.org/seinet/ |
| | VertNet | https://vertnet.org |
| Interactions | Global Biotic Interactions (GloBI) | https://www.globalbioticinteractions.org |
| | Terrestrial Parasite Tracker | https://parasitetracker.org |

must create a free guest account to download data and customize most features. Access for logged-in users depends on their permissions at the level of both collections and user roles, thus allowing for fine-grained control over database administration and management.

The Arctos Collections home page (https://arctos.database.museum/home.cfm) provides details on each collection with a search link to its complete set of records. Collections data also may be accessed by selecting "Catalog Records" in the Search dropdown from the main menu. Users may opt to use the default search form, select from discipline-specific query presets, or customize the search interface with preferences that can be saved to the user's profile. In addition to searching catalog records, users can query data by Agents (people and organizations), Media, Places and Events, Publications and Projects, and Taxonomy. Sample collection searches for tissue samples used in a DNA barcoding study [53] and for 3D models created as part of the oMeso project (https://arctos.database.museum/project/10003438), respectively, are available on the Arctos website (https://arctosdb.org/about/quick-tour/quick-tour-search).

Users can view or visualize the results of a search in multiple ways. By default, results are viewed in a table format with a core set of fields. However, users may customize this output by adding additional fields to the display (including media if available) and by arranging the

columns in a preferred order. Downloads include all fields that are selected and displayed in the results set, and data are exported as a comma-separated file which can be viewed in any desktop or cloud-based application. Data can be mapped using BerkeleyMapper and results summarized as counts by a subset of fields. Additional options are available to logged-in users such as asynchronous downloads of particularly large or time-consuming requests as well as the ability to save searches with a URL to share with collaborators or other users.

Users also can access Arctos through an Applied Programming Interface (API) by obtaining an access key and by compliance with certain requirements (https://arctosdb.org/arctos-api-policy). In addition to employing the API to retrieve records, collections have started using it to develop their own customized search interface. For example, the Alabama Museum of Natural History has been working with groups of Management Information Systems (MIS) students since 2020 to develop a customized user interface for the Department of Museum Research and Collections, and is using Arctos' API to retrieve results under its own subdomain (https://arctos.museums.ua.edu/Search). This project highlights how Arctos tools can be applied for specific institutional needs and serves as an educational example of cross-disciplinary synergy.

Finally, each collection may license their data using one of seven Creative Commons (CC) licenses (https://creativecommons.org/share-your-work/cclicenses), which are downloaded along with data records. In addition, collections have the option of selecting an alternative license for exporting data to the Global Biodiversity Information Facility (the GBIF policy only accepts three CC licenses). Collections also may link to their own institutional policy that describe terms of use or may default to Arctos' Community Data Policy (https://arctosdb.org/arctosdata-policy). All of these options are part of the collections' metadata, which apply to the entire collections' dataset, and may be changed at any time. Separately, media also can be licensed using either a CC license or an institutional document. Unlike catalog records, media licensing is applied to each object individually.

## 4.2 Benefits of shared data and emphasis on data quality

The shared data structure in Arctos, combined with its breadth of data types and the diverse expertise of community members, promotes cross-disciplinary discovery, workflow efficiencies, and improved data quality. For example, shared taxonomic classifications benefit diverse collections that manage and use objects with different types of components (e.g., biological and cultural materials, Fig 6; fossils with minerals). As noted in section 3.2.4, collections can add a mixture of names to the identifications of these types of objects, thus allowing the records to be useful for ethnographic, geologic, and biodiversity research. Furthermore, distinctions can be made between "taxonomy of creation" (identification of animal or plant products that a cultural object is made of) and "taxonomy of use" (identification of organisms for which the object is used). One study [54] that cites Arctos specimens from the University of Alaska Museum's ethnology collection combined biological, statistical, ecological, and ethnographic data to examine how and why Northwest Coast halibut hooks used in Indigenous fishing have changed over ~150 years. By identifying these objects with both their cultural and biological names ("hook, halibut" and "*Hippoglossus stenolepis*" respectively; e.g., https://arctos.database.museum/guid/UAM:EH:UA64-021-0834), the data are more discoverable to researchers interested in halibut fishery management and/or its rich cultural history.

Shared localities with coordinate metadata in Arctos form a gazetteer of vetted data that is available to the entire community, thereby reducing redundant georeferencing effort (Fig 9). As a case study, the Museum of Vertebrate Zoology was able to match over 60% of ~2,000 new specimen records acquired from an orphaned bird collection to georeferenced localities

already present in Arctos. Shared data also enable novel discoveries about the contributions of people, organizations, and other parties within and beyond Arctos (Fig 5). Biographical and statistical information in Arctos produces a holistic view of an individual's career-long activities across institutions rather than partitioning that information by institution. It also provides opportunities to discover that names associated with different collections are indeed the same person, thus allowing for reconciliation of name variants and corroboration of low-resolution identities while making the data richer and more complete. The ability to store and share biographical information about agents across collections is an important feature of Arctos, especially for cultural and archival collections.

Each Arctos record has a public Annotation function on its object detail page, where users can report data issues and provide feedback to the collections. In addition, Arctos provides shared services that check for data inconsistencies across this comprehensive repository. Common errors that plague many collections over time can be flagged, such as records that have nonsensical dates (e.g., identification dates before collecting dates, collection date before collector's birth date, etc.) or localities that fall outside of geographic boundaries. Other data quality tools check for incompleteness in records such as those with missing parts, unreciprocated relationships, ungeoreferenced localities, gaps in taxonomic hierarchies and catalog numbers, and missing GenBank links that cite Arctos records. While all annotated data and data quality flags are public, it is the responsibility of the collection to verify and update its records.

## 4.3 Connectivity and the Digital Extended Specimen as core principles

Connectivity is a core principle of the Digital Extended Specimen concept [12,13,55], which revolves around one specimen or object with a single collecting event and links to its derivatives such as gene sequences, CT scans, isotope data, publications, and media. Since its establishment, the Arctos data platform has been built on the extended specimen network concept – that is, displaying and making accessible everything that is known about an object and its relationships, interactions, or derivatives. Arctos creates a web of knowledge with deep connections between catalog records and derived or associated data using reliable published resources for globally shared information (Fig 10). This makes Arctos a central hub of collection-related data and tools for the exploration and visualization of biological, geological, and cultural diversity in novel ways. Focus on the Digital Extended Specimen promotes interdisciplinary research into functional traits [56] and biological interactions [39,57,58], provides a critical foundation for global conservation in response to changing environments [59,60], and reflects the role that "next-generation collections" [17] play in advancing science and society.

Although the Digital Extended Specimen is an important principle for modern-day collections, the reality is often more complicated. Specifically, an individual organism or object may be represented by multiple collecting events, preparators, preparations or preservations, and other metadata. Arctos discussions on how to address this challenge led to the creation of a community-wide "Arctos Entity" collection that combines component records sharing the same organism, object, or event ID into a single dashboard with a unique, shareable URL (S2 Fig). Components are linked using identifiers such as a bird band number or zoo Global Accession Number (GAN), and the Arctos Entity record connects its own inclusive records via individual Arctos URLs. The Arctos Entity collection has been used to connect three pieces of an 1840's dress, radio-tracked Golden Eagles (*Aquila chrysaetos*) that were banded and blood-sampled as nestlings, genetic samples of captive zoo animals taken at different times, and multiple events associated with repeat blood samples from Mexican wolves (*Canis lupus baileyi*) in a federal endangered species recovery program and later archived in a museum biorepository. Thus, Arctos Entity records provide a means of bridging data from living

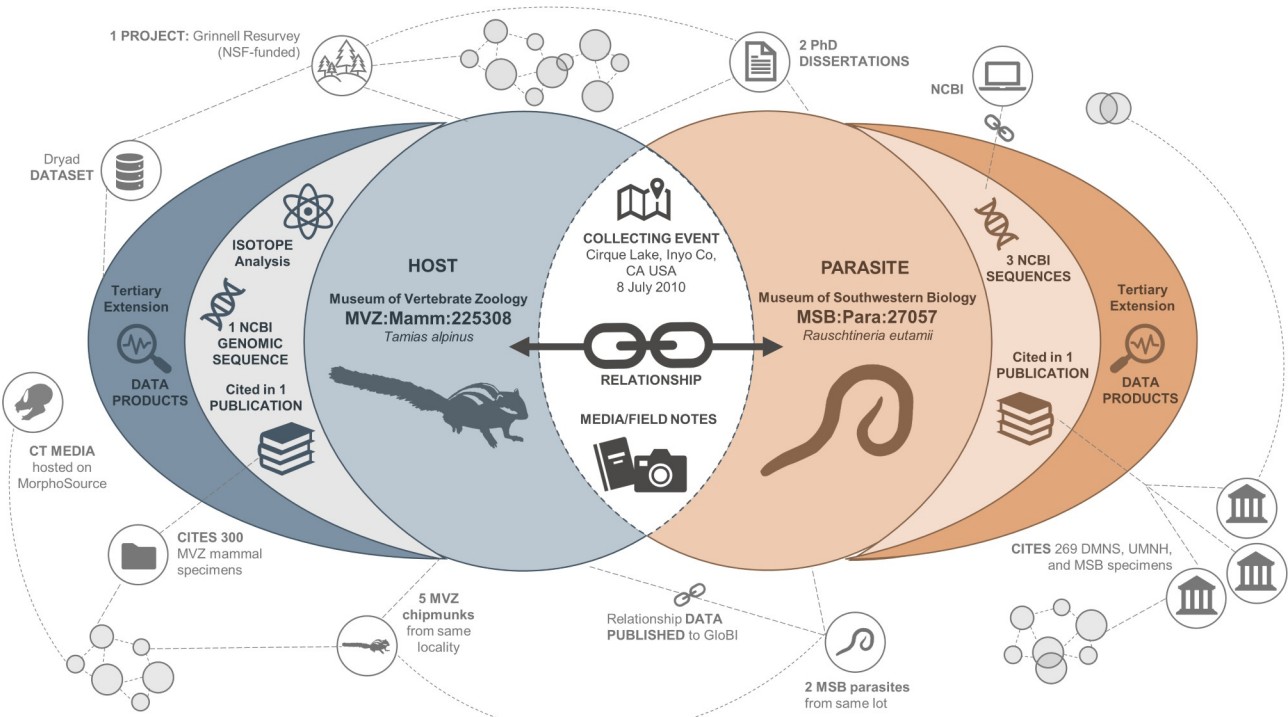

**Fig 10. Multidimensional extended specimens in Arctos.** Schematic illustration of two multidimensional extended specimens housed in different Arctos institutions (*Tamias alpinus*, MVZ:Mamm:225308; *Rauschtineria eutamii*, MSB:Para:27057) that share primary, secondary, and tertiary components and are directly related to each other. In 2010, researchers collected an Alpine Chipmunk (*Tamias alpinus*) along with a pinworm parasite (*Rauschtineria eutamii*) in Inyo County, California. These specimens are accessioned at separate institutions (Museum of Vertebrate Zoology, Museum of Southwestern Biology) but share host-parasite biotic interactions, collector, georeferenced locality, collection date, and primary source material such as field notes. Thus, any updates to host or parasite metadata (e.g., identification, locality, date) are reflected dynamically in both records. Together, the chipmunk and parasite were used in graduate student research producing two dissertations and at least two publications [57,60] that cited 569 specimens from four Arctos collections. The host is linked to one genomic sequence deposited at the National Center for Biotechnology Information (NCBI) Sequence Read Archive (SRA), and the parasite is linked dynamically to three GenBank sequences at NCBI. In addition, the chipmunk was one of 10,987 vertebrate specimens and observations collected in the southern Sierra Nevada as part of a statewide effort to resurvey California biodiversity (Grinnell Resurvey Project, https://arctos.database.museum/project/10000244), which resulted in 22 publications and 2,219 citations in Arctos. Furthermore, the host-parasite interactions have been harvested by the GLoBI platform [39] which publishes reciprocal links to the Arctos records. Thus, each specimen has its own extended data network with links, media, and publications that are searchable across collections and institutions. The extensive data associated with these records can be viewed directly through Arctos using the following URLs: https://arctos.database.museum/guid/MVZ:Mamm:225308; http://arctos.database.museum/guid/MSB:Para:27057. This example illustrates how the sharing of information facilitates data management and promotes interdisciplinary study.

collections (e.g., in zoos) and preserved material in museums [20]. Entity identifiers are passed to biodiversity data publishers via the Darwin Core organismID field, enabling aggregator portals and users to resolve these different records as the same individual. The flexibility of this model allows additional samples and observations to be continually linked to their Arctos Entity record as more data and samples are recorded.

## 4.4 Arctos as an educational resource

From its beginnings, Arctos has spearheaded collection-based inquiries for undergraduate education because of its web accessibility and richly linked data [3]. Students interested in biodiversity, evolutionary dynamics, spatiotemporal variation, cultural heritage, responses to anthropogenic change, and other topics have access to an array of data and tools that can initiate and answer interdisciplinary questions. This is exemplified by educational platforms where Arctos-based modules are posted for open-access class exercises (e.g., Aim-UP! https://arctos.

database.museum/project/10003944; [3]). Arctos also is used as a live classroom tool at universities (e.g., "Natural History Museums & Biodiversity Science" at the University of California Berkeley; "Museum Practicum in Advanced Collections Management" at the University of Colorado Boulder; "Mammalogy" at the University of New Mexico; "Fundamentals of Museum Studies I" at the University of Alaska Fairbanks) and has been important in training undergraduate and graduate students, post-baccalaureates, and postdoctoral researchers in museum curation and data management [61]. Additionally, collection staff have used Arctos to creatively integrate museum objects with artwork and public engagement in an effort to educate students and the broader community about collections. For example, the Alabama Museum of Natural History collaborated with the University of Alabama Fashion Archive to host a colorfest on social media that invited the public to interact with museum objects in art projects (#ColorOurCollections, http://library.nyam.org/colorourcollections; https://arctos. database.museum/project/10003310). In another art-based public exhibit, students, volunteers, and researchers spent a semester at the University of Wyoming Museum of Vertebrates creating original art pieces inspired by natural history objects, which were then displayed in a public show at the Berry Biodiversity Conservation Center (Art Inspired by Life, https://arctos. database.museum/project/10002855). At the University of Alaska Museum, staff in the Archaeology and Ethnology & History Departments work collaboratively with local, regional, national, and international organizations to highlight cultural items and Alaska Native heritage [62]. As the Arctos network expands, so will its educational role in promoting awareness of the rich legacy and potential of museum collections.

## 5. Discussion

The digitization era has prompted the powerful mobilization of natural history data, bringing both opportunities and challenges [4]. As a result, numerous collection management systems have evolved to meet the growing infrastructure needs for supporting large scale data, and these are well-summarized by iDigBio [18]. In a sea of options, we believe that Arctos is singular in its ability to dynamically connect data in a richly annotated ecosystem and communicate this information across the semantic web to realize the Digital Extended Specimen concept [13]. Arctos has a long track record as a model for "next-generation collections" and associated interdisciplinary research that addresses current and future societal challenges [17,63–65]. Central to this connectivity is the Arctos community, a network of collections professionals that hold the shared vision of providing highly standardized, deeply linked, and accessible data. By uniquely committing to hosting and managing data in a shared environment and collaborating on database-wide decision-making, Arctos members participate in collective data improvements and efficiencies to further the Arctos mission of promoting high quality data that empower research, education, and conservation to ultimately address pressing global challenges [17,66]. In addition to supporting digital records and data products, Arctos facilitates the management of the physical collections underpinning these data through its multiple collection management tools and its capability to document preservation and condition status through time. Therefore, Arctos provides support for collections stewards who organize, manage, and archive biological, geological, and cultural heritage collections for future generations through integrating information about physical and digital collections within a single architecture.

### 5.1 Financial support and long-term sustainability

Arctos' longevity of nearly three decades may be due in part to its nimble development model for addressing novel needs and information types in response to changing technology,

workflows, ethical considerations, and regulations [31,67–69]. Thus far, we have focused on the community and technical aspects of the Arctos database platform and its core development principles of standardization, flexibility, interdisciplinarity, and connectivity. However, maintaining these accessible and managed technologies ultimately is premised on reliable funding and dedicated staff. Unfortunately, museum staff are too often overextended with diverse responsibilities and limited financial resources [66]. Despite the fundamental importance of these digital resources for natural and cultural diversity, financial sustainability remains an ongoing concern for the global museum community [70].

Arctos prioritizes balancing sustainability with affordability for cash-strapped museums, and thus has relied on a mixture of volunteer and paid efforts in a scalable financial model. Currently Arctos is supported financially by subscription or membership fees from its member institutions, supplemented by external grants and in-kind support in the form of personnel subsidized by its members (see section 3.1 on Community). Membership fees are on a sliding scale based on collection size and ability to pay, and fee waivers are granted to a small number of collections that lack funding support (see https://arctosdb.org for details). Membership entitles each collection, regardless of size or contribution, to all benefits of Arctos including data hosting at TACC, community discussion boards, mentors, programming support, help resources such as documentation and tutorials, data backup and security, and publishing to external data aggregators and platforms. Collections that have specific development needs may need to provide additional institutional or grant-funded support, and grant-funded projects are given high priority. Existing or prospective members who are submitting grant proposals should consult with the director of Executive Officers and with the technical staff to obtain a letter of support (if necessary) and to ensure that the work, timeline, and budget are appropriate.

Starting in 2021, the Arctos community sought a more scalable, stable financial model for its growing consortium of independent and diverse institutions and initiated an internal task force to focus on the problem. The task force recognized that diversifying income sources are key to long-term sustainability [65,71] but that costs for administrative overhead (i.e., payroll, filing taxes, legal support) were daunting, especially for Arctos officers who are biologists and cultural historians with minimal experience in business or nonprofit administration. While US nonprofit status (i.e., 501(c)(3) of the US tax code) was attractive and seemed like an intuitive fit since most of the Arctos institutions also are nonprofit organizations, establishing and maintaining the administration of such an organization was financially prohibitive. Instead, the task force concluded that fiscal sponsorship by an existing nonprofit organization provided the most practical and ideal solution. Fiscal sponsorship is increasingly more common in the United States and refers to the arrangement whereby a nonprofit organization supports "projects" with similarly aligned missions by providing an array of administrative support including legal advice, human resources, payroll services, accounting, etc. in exchange for an operational overhead fee (e.g., a percentage of funds received by the project [72]). Sponsored projects retain their governance and operational structure, oversee their own strategic planning, and are ultimately responsible for paying their operational costs (e.g., staff salaries). It is not uncommon for fiscally sponsored projects to use this status as a stepping stone to independent nonprofit status in the future, but the immediate benefits are clear: sponsor-borne administrative costs, legal and other consulting support, and conferral of tax-exempt status. The latter allows Arctos to accept private donations and pursue other kinds of grants aimed at nonprofit organizations (i.e., funding sources outside of traditional agencies), which is part of diversifying the Arctos income beyond membership fees.

In 2022, Arctos became fiscally sponsored by the nonprofit organization Community Initiatives (https://communityinitiatives.org). This sponsorship and subsequent transition to

nonprofit status led to the establishment of more formal procedures for Arctos member institutions, created necessary user and privacy policies, and resulted in legal review and updating of the Memorandum of Understanding between Arctos and member institutions. The Arctos task force chose Community Initiatives because they offer a comprehensive suite of services and, importantly, have a track record of supporting projects like Arctos, i.e., a network of science or academically focused institutions with core values in research and accessibility for the public good.

The final component of sustainability is establishing key strategic partnerships. Arctos accomplishes its many functions and goals with external partners that include TACC for database hosting and infrastructure support, and VertNet for mobilizing records to aggregator portals such as GBIF and iDigBio. Arctos' earliest partnership includes the Berkeley Natural History Museums consortium at UC Berkeley, which provides database administration support, programming expertise, and services such as the embedded BerkeleyMapper (spatial data viewer). Partnerships with other data platforms have led to scalable, sustainable development innovations - including GenBank, Global Biotic Interactions [39], Global Genome Biodiversity Network [73,74], Global Names Architecture [32], IsoBank [40,41], Morphosource [42], and World Register of Marine Species [34]. Importantly, Arctos has collaborated with experts from different communities to establish metadata standards and best practices that foster digital connections for collection records; early in its history, for example, Arctos worked with GenBank to pioneer reciprocal relationships between specimens and DNA sequences using Darwin Core Triplets in the specimen_voucher field. Collaborative databasing has proven to be a model of sustainability by sharing expertise, encouraging efficiencies, and promoting compatibility [15,34,75].

## 6. Conclusion

The strength of Arctos lies in its well-established, growing community of museum professionals who oversee and shape the Arctos experience, and in its efforts to remain at the forefront of collection management practices in a connected digital ecosystem. Community-led innovations, diverse expertise and perspectives, data standardization, and a flexible development model make Arctos practical for managing collections of all sizes and types. Users gain mentorship and community support while having access to a suite of tools that facilitate management of data about both physical and digital objects. As a community of practice and a data platform, Arctos aims to rigorously document records that are discoverable, linked, and secure while ensuring the accessibility of collections and their holdings for the benefit of society.

## Supporting information

**S1 Fig. Schematic diagram for Arctos.** Schematic diagram highlighting data that are shared and curated across collections versus data that are controlled by individual collections. Some data can be collection-controlled or community-curated depending on the collection's preference. Solid lines directly link data to catalog records, while dotted lines represent other relationships between tables. Certain shared data (e.g., localities) can be locked from editing by another user. Full details on all Arctos data tables and their structure are publicly available through the portal (https://arctos.database.museum/tblbrowse.cfm).
(TIF)

**S2 Fig. Example of the Arctos Entity model.** The Arctos Entity model links diverse catalog records to a single unifying record for increased discoverability. Here, Record A is a bird observation with an associated blood sample, Record B is a second observation of the same

individual with associated radio telemetry data, Record C is the vouchered specimen, and Records D and E are endoparasites and ectoparasites, respectively, taken from the specimen. Three examples illustrate how the Entity record functions in Arctos to compile and unite multiple related occurrences or records of a single organism or collection object under one persistent identifier (Arctos base URL combined with the Darwin Core Triplet for the Entity record): (1) https://arctos.database.museum/guid/Arctos:Entity:16 links the cataloged blood sample of a Golden Eagle (*Aquila chrysaetos*) chick banded at its nest in 2014 (MVZ: Bird:193216) with data from a radio transmitter device that tracked the individual's last known location to Mexico in 2017; that observation was cataloged in Arctos as MVZObs:Bird:4792. The coordinates for the original sampling locality are encumbered to protect the eagle nest. (2) A single endangered Mexican wolf (*Canis lupus baileyi*; https://arctos.database.museum/guid/Arctos:Entity:134) was monitored through a federal conservation program with regular blood sampling at different times in two zoos and the Wolf Management Facility, Sevilleta National Wildlife Refuge (MSB:Mamm:341613, MSB:Mamm:231704). Once moribund, the entire specimen was preserved and cataloged as MSB:Mamm:341614. (3) A captive Sumatran Orangutan (*Pongo abelii*; https://arctos.database.museum/guid/Arctos:Entity:204) in the Albuquerque BioPark Zoo has had at least 20 blood samples taken between 2011 and 2020 that are preserved in different ways and archived at the Museum of Southwestern Biology.
(TIF)

## Acknowledgments

We thank all members of the Arctos Working Group for their unflagging efforts to improve Arctos and keep it an active, functioning, and engaged community and platform. We also thank the generations of undergraduate and graduate students, post-baccalaureates, collection managers, curators, technicians, and volunteers who perform daily collection tasks using Arctos at member institutions. The following individuals and collaborators have contributed invaluable expertise, perspectives, and support that have helped to enrich and expand Arctos as both a data platform and community: Stan Blum, John Deck, Jonathan Dunnum, Joyce Gross, Aren Gunderson, Steffi Ickert-Bond, Gordon Jarrell, Craig Moritz, Kyndall B.P. Hildebrandt, Barbara Stein, Lam Voong, Cam Webb, John Wieczorek; Global Biotic Interactions (GloBI; Jorrit Poelen); Global Genome Biodiversity Network (GGBN; Katharine Barker); Global Names Architecture (Dmitry Mozzherin); Integrated Digitized Biocollections (iDigBio; Gil Nelson, Deborah Paul, Erica Krimmel), and the Texas Advanced Computing Center (TACC; Chris Jordan). Finally, we thank Community Initiatives, especially Brandy Shah and Rose Cohen Westbrooke, for their guidance and expertise in our transition to fiscal sponsorship.

## Author Contributions

**Conceptualization:** Carla Cicero, Michelle S. Koo, Emily Braker.

**Visualization:** Carla Cicero, Michelle S. Koo, Emily Braker.

**Writing – original draft:** Carla Cicero, Michelle S. Koo, Emily Braker.

**Writing – review & editing:** Carla Cicero, Michelle S. Koo, Emily Braker, John Abbott, David Bloom, Mariel Campbell, Joseph A. Cook, John R. Demboski, Andrew C. Doll, Lindsey M. Frederick, Angela J. Linn, Teresa J. Mayfield-Meyer, Dusty L. McDonald, Michael W. Nachman, Link E. Olson, Dawn Roberts, Derek S. Sikes, Christopher C. Witt, Elizabeth A. Wommack.

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
