## [Decision Letter · Decision Letter 0]

12 Jan 2024

PONE-D-23-37198Arctos: Community-driven innovations for managing biodiversity and cultural collectionsPLOS ONE

Dear Dr. Cicero,

Thank you for submitting your manuscript to PLOS ONE. After careful consideration, we feel that it has merit but does not fully meet PLOS ONE’s publication criteria as it currently stands. Therefore, we invite you to submit a revised version of the manuscript that addresses the points raised during the review process.

**Arrange the manuscript so that also Material & Methods as well as Results are included;**

**Avoid repetition between Abstract and Introduction trying to cut text when it does not add particular value to the rich info already provided;**

**Provide some example outcomes of files / data eventually as supplementary material;**

**Offer more details on financial sustainability of Arctos and how it compares / integrates with already existing platforms.**

We look forward to receiving your revised manuscript.

Kind regards,

Carlo Meloro

Academic Editor

PLOS ONE

Journal Requirements:

We thank all members of the Arctos Working Group for their unflagging efforts to improve Arctos and keep it an active, functioning, and engaged community and platform. We also thank the generations of undergraduate and graduate students, post-baccalaureates, collection managers, curators, and technicians who perform daily collection tasks using Arctos at member institutions. The following individuals and collaborators have contributed invaluable expertise, perspectives, and support that have helped to enrich and expand Arctos as both a data platform and community: Stan Blum, John Deck, Jonathan Dunnum, Joyce Gross, Steffi Ickert-Bond, Gordon Jarrell, Craig Moritz, Kyndall B.P. Hildebrandt, Barbara Stein, Lam Voong, Cam Webb, John Wieczorek; Global Biotic Interactions (GloBI; Jorrit Poelen), Global Genome Biodiversity Network (GGBN; Katharine Barker), Integrated Digitized Biocollections (iDigBio; Gil Nelson, Deborah Paul, Erica Krimmel), and the Texas Advanced Computing Center (TACC; Chris Jordan). We thank the National Science Foundation for funding specific to the development and sustainability of Arctos (DBI-9630909, DBI-9876837, DEB-9981915, DBI-2034593, DBI-2034568, DBI-2034577), as well as the Robert & Patricia Switzer Foundation for awarding Arctos a Leadership Grant in 2023; additional grants from various sources have funded collection-specific initiatives that resulted in Arctos improvements. Finally, we thank Community Initiatives, especially Brandy Shah and Rose Cohen Westbrooke, for their guidance and expertise in our transition to fiscal sponsorship.

Additional Editor Comments:

This is a well written paper that present the benefits of Arctos. I like the paper structure although I must say that I dislike repetition that in the current status of the article are present quite a lot. This occurs especially in between the Abstract and the Introduction so I recommend to change one or the other.

Another thing that should be better implemented is the structure into Methods and Results / Discussion section. A good example you can try to follow is this:

The ROCEEH Out of Africa Database (ROAD): A large-scale research database serves as an indispensable tool for human evolutionary studies

https://journals.plos.org/plosone/article?id=10.1371/journal.pone.0289513

I do not think that you need some extra analyses or statistics to be added to the paper, you just need to organise them a little bit in a more effective way. For instance, based on your map it seems that your database deals with collections worldwide although you got mainly 4 countries that contribute to it.

It would be good to clarify the geographic scale of this impressive database that at the moment is quite rich in details [which is at the same time scary and exciting for the first time users]. It would be good to have examples of search tools and effective way to extract data using .csv format or .txt because the amount of info included in Arctos is quite a lot and arranged as text with multiple outcomes (let’s say a sort of guidelines to optimise specimen/collection searches).

Last but not least, it will be important to clarify the staff time effort and financial scheme a bit more in details to show how sustainable and replicable this system is.

Below I provide some detailed comments that I hope you will be able to address:

The use of the word biological should be broaden in the context of including also geology and mineralogy...perhaps with "natural" or eventually add the world "geological" to biological. This appears already in the abstract and is repeated in the Introduction

The first part of the Intro 66-77 is too similar to the abstract or the other way round. Try to change on or the other to avoid repetition. I suggest to change the abstract and make it a bit more efficient in summarising the strengths of Arctos

Line 88: what FAIR stands for? Move its definition in parenthesis straight after the term FAIR and not in line 89

Line 92-97: this sounds like a repetition of what you introduced before, make it shorter as a single sentence just before introducing ARCTOS

Line 97-1-4: again this is the same as in the Abstract, so please change one or the other

Line 128: what do you mean for "archives"? do you mean written archive records or more specific to some of the museum object records?

Line 131: nice but it would be good to see some good examples from the literature...any example ref. showing the productive scientific dialogue Arctos introduced?

Line 181: this is a bit repetitive now and it would be good if you could add some numbers to these data. How many collector managers? how many researchers? educators? and so on...but more specifically what is their role / responsibility?

Line 188: nice, does this mean that Arctos will be ideally also open to other part of the world or is specifically dealing with US federal government only?

Line 204-212: all good here but I wonder how the community comes together? Through the web? do you do regular zoom meeting? who is in charge of organising them? I assume there is still some dedicated personnel that check Arctos functionality / web security and so on...this should be specified a bit. There must be IT personnel dedicated to Arctos...how many and how much does that cost? It would be good to know for anyone interested in joining or developing something similar. Also, what are the responsibility of anyone that decides to join? You answer this a bit in the next section but it is not clear how many members each panel has and where are they based.

Is Arctos management part of their "job role"? In UK for instance the NHM collection managements are pretty much busy and the NHM data portal (https://data.nhm.ac.uk/) is possibly curated by extra additional / time / staff / volunteers. How much time is that and what is the working force involved for Arctos?

In Fig. 2 would be good to figure out who are the 4 countries involved and if there is aim and scope to expand this geographically

The colour of the frequency bar is ok but seems trivial to understand. If this is just a nice visualisation perhaps use colour gradients across blue or always the same colour (the first orange bar looks quite odd)

In Fig. 3 it will be good to know how many people per committed are present

Line 283: is there any example of digital records that are easy to find. Let's say I want to find in Arctos all the 3D models of vertebrate skull. Is it possible to find it? Perhaps some screenshot example on how to find this kind of record or on where it is registered would be good to show. Also I wonder if there is a double check system on the records that might be biased or inaccurate or might not correspond to original recording (e.g. sometimes collections change acronym or catalogue numbers that used to be written in paper). Any record of catalogues / or change in specimens code or way to correct / update an old record?

Line 491: I like this example but the figure associated to it is quite conceptual...it will be good to have maybe as a supplementary file the whole record in pdf associated to these specimens via the database.

Same stands for maps / records ....some examples that can help the users to do an effective search. I tried some functionality myself and realised that the amount of info is massive so you need to be quite focused when searching a particular item.

Line 590 onwards: I think you need to be more specific here. Who are the partners interested in financing Arctos and how sustainable over the long term this is going to be? In short readers will be interested to understand if your proposed Arctos database system can be joined inn or re-created for other museums by paying attention to financial constraints and time that museum staff need to work on this.

How much average time a museum curator must dedicate to ensure Arctos will survive over long term? How many "permanent" dedicated workers are needed to take care of Arctos? and most importantly: what sort of cyber security is applied and how much does that cost?

Reviewers' comments:

Reviewer's Responses to Questions

**Comments to the Author**

1. Is the manuscript technically sound, and do the data support the conclusions?

Reviewer #1: Yes

Reviewer #2: No

2. Has the statistical analysis been performed appropriately and rigorously? 

Reviewer #1: Yes

Reviewer #2: N/A

3. Have the authors made all data underlying the findings in their manuscript fully available?

Reviewer #1: Yes

Reviewer #2: No

4. Is the manuscript presented in an intelligible fashion and written in standard English?

Reviewer #1: Yes

Reviewer #2: Yes

5. Review Comments to the Author

Reviewer #1: Thank you very much for allowing me to review your manuscript Arctos: Community-driven innovations for managing biodiversity and cultural collections. It can be challenging to describe database systems effectively, and this paper achieves this, although there are a few suggestions for improvement.

Major comments

------------------

The abstract establishes the objective of this paper to highlight the unique aspects of Arctos. However, no reference is made to other leading collection management systems (Specify; KE EMu). A comparison with these systems is required to ground-truth this claim of uniqueness. For example, the paper claims a key unique feature of Arctos is its ability to connect with external repositories, but is similar functionality not provided by Specify and KE EMu?

A comparison of the Arctos model with other collection management systems would be helpful as part of establishing the difference between the systems. Existing comparisons exist [1], but the paper should discuss how the Arctos model can best support the information requirements of collections.

The section “A sustainable future for Arctos” needs clarification. The authors assert that this model was implemented in 2022, in which case some results should be included. Have non-profit organisations been found to provide fiscal support and establish a long-term future for Arctos? Have the resource constraints mentioned been solved? If it is the case that non-profit organisations have been providing support but cannot be named for contractual reasons, this should be made explicit.

Data licensing and privacy information would also be useful, particularly with Arctos’ capability to share data entities across collections. If this sharing is cross-institutional, does the use of the system require institutions to permit the reuse of their data? Does the system allow data embargoes? The paper describes Artos’ ability to “store and share biographical information about agents across collections,” which has user privacy implications that should be addressed.

Minor comments

-------------------

The paper describes the database model, but a schematic diagram would aid understanding of this model.

A few additional citations are needed:

“Lucee-based web interface” (page 7)

“Apache 2” (page 7)

The paper describes the user community well, but it would be interesting to mention the development community. As this project is open-source, are there community or other technical contributions?

References

------------

[1] https://www.idigbio.org/wiki/images/4/4a/Brenskelle-iDB2.pdf

Reviewer #2: I suggest you substitute "natural" for "biological" throughout the paper, since geological material (rocks, minerals, fossils, but also cores, seismic records, etc) are also abundantly present in collections and they are neither biological nor cultural. The paper would also benefit from a very brief "Conclusion" paragraph, just after "A sustainable future for Arctos", where you would summarize and wrap up the advantages of Arctos.

6. PLOS authors have the option to publish the peer review history of their article (what does this mean?). If published, this will include your full peer review and any attached files.

Reviewer #1: No

Reviewer #2: No

---

## [Author Response · Author response to Decision Letter 0]

28 Mar 2024

Thank you for submitting your manuscript to PLOS ONE. After careful consideration, we feel that it has merit but does not fully meet PLOS ONE’s publication criteria as it currently stands. Therefore, we invite you to submit a revised version of the manuscript that addresses the points raised during the review process.

Response: Thank you for this comment. We have addressed all of the points raised during the review process. See below.

Arrange the manuscript so that also Material & Methods as well as Results are included;

Response: We have done a major revision so that the manuscript now includes sections for Methods, Results, Discussion, and Conclusions. We reorganized text so that it is in the appropriate section.

Avoid repetition between Abstract and Introduction trying to cut text when it does not add particular value to the rich info already provided;

Response: We have modified both the Abstract and Introduction to minimize repetition, and also checked for repetition throughout the manuscript.

Provide some example outcomes of files / data eventually as supplementary material;

Response: We have added three supplementary files as example data outputs. We also added a supplementary table that provides links to different kinds of attributes that are an integral part of Arctos searches and outcomes.

Offer more details on financial sustainability of Arctos and how it compares / integrates with already existing platforms.

Response: We moved all of the financial information including subscription fees under the Level 2 heading “Financial support and long-term sustainability,” and provide more details including what it means to be under fiscal sponsorship. With regard to other platforms, we discuss Arctos’ integration with external platforms in detail but those integrations do not affect the sustainability of Arctos itself. Furthermore, the goal of this paper is not to compare Arctos to other collection management systems but rather to describe the capabilities of Arctos. Thus, it is outside the scope to compare our financial model with other systems. However, we reference the survey by iDigBio on different collection management systems and their capabilities, and readers can refer to those sites for more information about other systems.

If applicable, we recommend that you deposit your laboratory protocols in protocols.io to enhance the reproducibility of your results. Protocols.io assigns your protocol its own identifier (DOI) so that it can be cited independently in the future. For instructions see: https://journals.plos.org/plosone/s/submission-guidelines#loc-laboratory-protocols. Additionally, PLOS ONE offers an option for publishing peer-reviewed Lab Protocol articles, which describe protocols hosted on protocols.io. 

Response: This is not applicable to our manuscript.

Journal Requirements:

Response: We have followed the PLOS ONE style requirements including for file naming.

2. We note that you have provided funding information that is not currently declared in your Funding Statement. However, funding information should not appear in the Acknowledgments section or other areas of your manuscript. We will only publish funding information present in the Funding Statement section of the online submission form. Please remove any funding-related text from the manuscript and let us know how you would like to update your Funding Statement. Currently, your Funding Statement reads as follows: The author(s) received no specific funding for this work.

Response: We have removed the funding information from the Acknowledgements section. Funding information does not appear elsewhere in the text. We have added an amended Funding Statement to the cover letter which we repeat here:

Funding Statement: Arctos has received the following grants from the National Science Foundation for development and sustainability: DBI-9630909, DBI-9876837, DEB-9981915, DBI-2034593, DBI-2034568, DBI-2034577. Additional funding from various sources have supported collection-specific initiatives that resulted in Arctos improvements. The Robert & Patricia Switzer Foundation awarded Arctos a Leadership Grant in 2023.

Additional Editor Comments:

This is a well written paper that present the benefits of Arctos. I like the paper structure although I must say that I dislike repetition that in the current status of the article are present quite a lot. This occurs especially in between the Abstract and the Introduction so I recommend to change one or the other.

Response: See comment above regarding revision of Abstract and Introduction to remove repetition. We have also minimized repetition throughout the manuscript.

Another thing that should be better implemented is the structure into Methods and Results / Discussion section. A good example you can try to follow is this:

The ROCEEH Out of Africa Database (ROAD): A large-scale research database serves as an indispensable tool for human evolutionary studies

https://journals.plos.org/plosone/article?id=10.1371/journal.pone.0289513

Response: See comment above. We used this excellent example to rework the manuscript structure and rearranged the text into sections on Methods, Results, and Discussion. We also added a Conclusions section.

I do not think that you need some extra analyses or statistics to be added to the paper, you just need to organise them a little bit in a more effective way. 

Response: We have completely reorganized the paper (see response above) to make a more effective presentation.

For instance, based on your map it seems that your database deals with collections worldwide although you got mainly 4 countries that contribute to it.

It would be good to clarify the geographic scale of this impressive database that at the moment is quite rich in details [which is at the same time scary and exciting for the first time users].

Response: We updated Fig 2 with the current number of countries (7) that have an institution using Arctos as a database platform, and also provided more information in the text on the geographic scope of objects represented by a record in Arctos. 

It would be good to have examples of search tools and effective way to extract data using .csv format or .txt because the amount of info included in Arctos is quite a lot and arranged as text with multiple outcomes (let’s say a sort of guidelines to optimise specimen/collection searches).

Response: We added a new Level 2 section under Results (Access, customization, and licensing) that provides more information on how to get data out of Arctos. We also added two examples of how to search data and the resulting outcomes (supplementary files S1 and S2). 

Last but not least, it will be important to clarify the staff time effort and financial scheme a bit more in details to show how sustainable and replicable this system is.

Response: We have expanded the sustainability section to provide more details on Arctos’ funding support, the financial model, and our strategic partnerships. We have also clarified the composition of the community (under Methods and caption to Fig 3) and indicate paid staff vs volunteers. Pure replication of this system would be difficult because one would have to replicate both the database platform and the community as the two are closely intertwined. Nonetheless, as stated in the brief history section (Level 2 heading under Introduction), a separate instance of Arctos is established at the Museum of Comparative Zoology (although the two systems are not in sync with regard to code).

The use of the word biological should be broaden in the context of including also geology and mineralogy...perhaps with "natural" or eventually add the world "geological" to biological. This appears already in the abstract and is repeated in the Introduction.

Response: Thank you for the suggestion. We have modified the text to make it clear that we are talking about “biological, geological, and cultural” collections data and diversity. We have also changed the title of the paper so that it now refers to “natural and cultural history collections.”

The first part of the Intro 66-77 is too similar to the abstract or the other way round. Try to change on or the other to avoid repetition. I suggest to change the abstract and make it a bit more efficient in summarising the strengths of Arctos

Response: We have made this change.

Line 88: what FAIR stands for? Move its definition in parenthesis straight after the term FAIR and not in line 89

Response: We removed this acronym and its definition from the text and made a more general statement about principles with citation that references the FAIR principles.

Line 92-97: this sounds like a repetition of what you introduced before, make it shorter as a single sentence just before introducing ARCTOS

Line 97-1-4: again this is the same as in the Abstract, so please change one or the other

Response: We have revised the text to reduce repetition.

Line 128: what do you mean for "archives"? do you mean written archive records or more specific to some of the museum object records?

Response: We added a description of what we mean by “archives” in Table 1.

Line 131: nice but it would be good to see some good examples from the literature...any example ref. showing the productive scientific dialogue Arctos introduced?

Response: We include a specific example of cross-disciplinary collaboration in the caption to Fig 10 (see also S3 File). The example cites a paper on host-parasite relationships that used specimens in multiple collections at different institutions, and also cites partnership with the Global Biotic Interactions (GloBI) platform to facilitate such research.

Line 181: this is a bit repetitive now and it would be good if you could add some numbers to these data. How many collector managers? how many researchers? educators? and so on...but more specifically what is their role / responsibility?

Response: We modified the text to indicate the number of collection-based operators who are directly responsible for day-to-day database activities. We cannot provide specific numbers by job title but we made their roles/responsibilities more clear. 

Line 188: nice, does this mean that Arctos will be ideally also open to other part of the world or is specifically dealing with US federal government only?

Response: We have added a sentence to clarify that there are no geographic or administrative restrictions.

Line 204-212: all good here but I wonder how the community comes together? Through the web? do you do regular zoom meeting? who is in charge of organising them? I assume there is still some dedicated personnel that check Arctos functionality / web security and so on...this should be specified a bit. There must be IT personnel dedicated to Arctos...how many and how much does that cost? It would be good to know for anyone interested in joining or developing something similar. Also, what are the responsibility of anyone that decides to join? You answer this a bit in the next section but it is not clear how many members each panel has and where are they based.

Is Arctos management part of their "job role"? In UK for instance the NHM collection managements are pretty much busy and the NHM data portal (https://data.nhm.ac.uk/) is possibly curated by extra additional / time / staff / volunteers. How much time is that and what is the working force involved for Arctos?

Response: We have revised the text under Community and the Fig 3 caption to clarify the roles of different people or groups within the community, expectations of Arctos community members, and community organization. 

In Fig. 2 would be good to figure out who are the 4 countries involved and if there is aim and scope to expand this geographically

Response: We updated this number from 4 to 7 in the figure and specified the countries in the text of the manuscript. We also added a sentence that “further international expansion is anticipated and welcome.”

The colour of the frequency bar is ok but seems trivial to understand. If this is just a nice visualisation perhaps use colour gradients across blue or always the same colour (the first orange bar looks quite odd)

Response: We have updated the figure to display a color gradient across the frequency bars rather than separate colors.

In Fig. 3 it will be good to know how many people per committed are present

Response: We have expanded the caption for Fig 3 to clarify the composition of the Arctos community and the roles of different individuals or groups. Because the numbers in each category are not constant, we felt that it would be misleading to add the number of people to the figure caption. However, a link to the Arctos website (https://arctosdb.org/contacts) provides a dynamic list of the individuals who serve Arctos in the different capacities.

Line 283: is there any example of digital records that are easy to find. Let's say I want to find in Arctos all the 3D models of vertebrate skull. Is it possible to find it? Perhaps some screenshot example on how to find this kind of record or on where it is registered would be good to show. 

Response: We have added a new section under Results (Access, customization, and licensing) that describes the basic procedures for conducting searches and viewing results. We also reference S2 File that provides an example of how to search for 3D models (not specifically for skulls although we note how to do that in the caption).

Also I wonder if there is a double check system on the records that might be biased or inaccurate or might not correspond to original recording (e.g. sometimes collections change acronym or catalogue numbers that used to be written in paper). Any record of catalogues / or change in specimens code or way to correct / update an old record?

Response: We have added a section on identifiers under Methods that summarizes how Arctos deals with old catalog numbers and other related information for a record. The section on Features discusses the availability of tools for correcting/updating records, among other things.

Line 491: I like this example but the figure associated to it is quite conceptual...it will be good to have maybe as a supplementary file the whole record in pdf associated to these specimens via the database.

Response: Thank you for the suggestion. We have added a supplementary file (S3 File) that shows Arctos screenshots of the mammal and related parasite records.

Same stands for maps / records ....some examples that can help the users to do an effective search. I tried some functionality myself and realised that the amount of info is massive so you need to be quite focused when searching a particular item.

Response: As noted above, we have added a new section under Results (Access, customization, and licensing) to make it clearer how to find and display data of interest. We give two examples of how to do specific queries and what those outcomes are, which we illustrate in supplementary files (S1 File and S2 File).

Line 590 onwards: I think you need to be more specific here. Who are the partners interested in financing Arctos and how sustainable over the long term this is going to be? In short readers will be interested to understand if your proposed Arctos database system can be joined inn or re-created for other museums by paying attention to financial constraints and time that museum staff need to work on this.

How much average time a museum curator must dedicate to ensure Arctos will survive over long term? How many "permanent" dedicated workers are needed to take care of Arctos? and most importantly: what sort of cyber security is applied and ho

---

## [Decision Letter · Decision Letter 1]

23 Apr 2024

Arctos: Community-driven innovations for managing natural and cultural collections

PONE-D-23-37198R1

Dear Dr. Cicero,

We’re pleased to inform you that your manuscript has been judged scientifically suitable for publication and will be formally accepted for publication once it meets all outstanding technical requirements.

Kind regards,

Carlo Meloro

Academic Editor

PLOS ONE

Additional Editor Comments (optional):

You did a good job in addressing all the main concerns of the paper.

Reviewers' comments:

Reviewer's Responses to Questions

**Comments to the Author**

1. If the authors have adequately addressed your comments raised in a previous round of review and you feel that this manuscript is now acceptable for publication, you may indicate that here to bypass the “Comments to the Author” section, enter your conflict of interest statement in the “Confidential to Editor” section, and submit your "Accept" recommendation.

Reviewer #1: All comments have been addressed

Reviewer #2: All comments have been addressed

2. Is the manuscript technically sound, and do the data support the conclusions?

Reviewer #1: Yes

Reviewer #2: Yes

3. Has the statistical analysis been performed appropriately and rigorously? 

Reviewer #1: Yes

Reviewer #2: N/A

4. Have the authors made all data underlying the findings in their manuscript fully available?

Reviewer #1: Yes

Reviewer #2: Yes

5. Is the manuscript presented in an intelligible fashion and written in standard English?

Reviewer #1: Yes

Reviewer #2: Yes

6. Review Comments to the Author

Reviewer #1: The changes do address my comments sufficiently. I get the point that a comparison with other similar systems is beyond the scope of this paper, but perhaps could be something for a future paper.

Reviewer #2: (No Response)

7. PLOS authors have the option to publish the peer review history of their article (what does this mean?). If published, this will include your full peer review and any attached files.

Reviewer #1: No

Reviewer #2: No
